# Landscape of essential growth and fluconazole-resistance genes in the human fungal pathogen *Cryptococcus neoformans*

R. Blake Billmyre [1,2,3,4*], Caroline J. Craig[4¤], Joshua W. Lyon[2,5], Claire Reichardt[1,2,3], Amy M. Kuhn[1,2], Michael T. Eickbush[4], Sarah E. Zanders[4,6]

1 Department of Genetics, Franklin College of Arts and Sciences, University of Georgia, Georgia, United States of America, 2 Department of Infectious Diseases, College of Veterinary Medicine, University of Georgia, Athens, Georgia, United States of America, 3 Department of Microbiology, Franklin College of Arts and Sciences, University of Georgia, Athens, Georgia, United States of America, 4 Stowers Institute for Medical Research, Kansas City, Missouri, United States of America, 5 Department of Pharmaceutical and Biological Sciences, College of Pharmacy, University of Georgia, Athens, Georgia, United States of America, 6 Department of Cell Biology and Physiology, University of Kansas Medical Center, Kansas City, Kansas, United States of America

¤ Current address: University of Utah Medical School, Salt Lake City, Utah, United States of America
* blake.billmyre@uga.edu

## Abstract

Fungi can cause devastating invasive infections, typically in immunocompromised patients. Treatment is complicated both by the evolutionary similarity between humans and fungi and by the frequent emergence of drug resistance. Studies in fungal pathogens have long been slowed by a lack of high-throughput tools and community resources that are common in model organisms. Here we demonstrate a high-throughput transposon mutagenesis and sequencing (TN-seq) system in *Cryptococcus neoformans* that enables genome-wide determination of gene essentiality. We employed a random forest machine learning approach to classify the *C. neoformans* genome as essential or nonessential, predicting 1,465 essential genes, including 302 that lack human orthologs. These genes are ideal targets for new antifungal drug development. TN-seq also enables genome-wide measurement of the fitness contribution of genes to phenotypes of interest. As proof of principle, we demonstrate the genome-wide contribution of genes to growth in fluconazole, a clinically used antifungal. We show a novel role for the well-studied *RIM101* pathway in fluconazole susceptibility. We also show that insertions of transposons into the 5′ upstream region can drive sensitization of essential genes, enabling screenlike assays of both essential and nonessential components of the genome. Using this approach, we demonstrate a role for mitochondrial function in fluconazole sensitivity, such that tuning down many essential mitochondrial genes via 5′ insertions can drive resistance to fluconazole. Our assay system will be valuable in future studies of *C. neoformans*, particularly in examining the consequences of genotypic diversity.

**Data availability statement:** Sequences have been deposited into the Sequence Read Archive under project accession no. PRJNA1134100. Original data underlying this manuscript is available as supplemental tables and supporting data files at 10.5281/zenodo.15264486. Data can also be accessed from the Stowers Original Data Repository at http://www.stowers.org/research/publications/libpb-2480, which also includes original plate images. Code can be found at 10.5281/zenodo.15297569.

**Funding:** We acknowledge support from NIH grants DP2GM132936 and R35GM151982 to SEZ. We acknowledge support from NIH grant DP2AI184725 to RBB and T32GM142623 to CER. We also acknowledge institutional support from the Stowers Institute to SEZ and from the University of Georgia to RBB. The funders had no role in the study design, data collection and analysis, decision to publish, or preparation of the manuscript.

**Competing interests:** The authors have declared that no competing interests exist.

**Abbreviations:** Ac, activator; DAmP, decreased abundance by mRNA perturbation; Ds, dissociation; ES, essentiality score.

## Introduction

Human fungal pathogens are a serious human health risk, causing approximately 3.8 million deaths a year [1]. Fungal infections are frequent in immunocompromised patients, particularly in conjunction with the HIV/AIDS epidemic. While the number of severely immunocompromised HIV patients has declined in recent years with the advent of HAART therapy, HIV management remains difficult in resource-limited areas and HIV-related illness remains the leading cause of death in sub-Saharan Africa. Even in resource-rich areas, HIV is still a significant challenge. In the US, up to 1 in 13 individuals who were at one point virally suppressed experience viral rebound [2]. Other sources of immunocompromise are also on the rise as solid organ transplants and the accompanying immunosuppression become more common. In combination, we face continued threats from fungal diseases.

In addition, treating fungal infections is highly challenging. On an evolutionary scale, fungal pathogens are more closely related to their human hosts than almost every other infectious disease, except animal pathogens like parasitic worms. This high relatedness makes fungi strong tractable models for human biology, as many aspects of mammalian biology are conserved in fungi. A drawback of this conservation is that many potential inhibitors of fungal growth are unacceptably toxic to humans because they inhibit shared biological pathways [3]. As a result, we have only five classes of antifungal drugs available to treat invasive diseases: azoles, echinocandins, polyenes, the nucleoside analog 5FC, and a relatively new drug ibrexafungerp. Two of these five classes, echinocandins and ibrexafungerp, target the same gene (*FKS1*). Most of these drugs target biology unique to fungi. For example, azoles, such as fluconazole, inhibit the essential activity of Erg11, which is responsible for part of the biosynthesis of ergosterol, a fungal analog of cholesterol not found in humans. The paucity of antifungal drugs means that when resistance emerges or when a species is intrinsically resistant to an antifungal drug, there are few therapeutic alternatives.

Among the most dangerous fungal pathogens are species from the *Cryptococcus* pathogen species complex [4]. *Cryptococcus neoformans* is the most common source of infections within a pathogenic species complex composed of at least 7 different species of pathogens [5]. The vast majority of the approximately 147,000 annual deaths from Cryptococcosis [1] are caused by infections with *C. neoformans* and occur in sub-Saharan Africa in conjunction with the HIV/AIDS epidemic [1]. *C. neoformans* rarely infects immunocompetent individuals. In immunosuppressed individuals, infection typically begins in the lungs as fungal pneumonia and can then disseminate through the bloodstream into the brain to cause fungal meningitis. Cryptococcal meningitis is uniformly fatal if left untreated.

Treating Cryptococcal meningitis is complicated by the fact that *C. neoformans* is intrinsically resistant to both echinocandins and ibrexafungerp [6,7]. Multidrug resistance is not common because invasive fungal pathogens, including *C. neoformans*, are generally acquired from environmental reservoirs. Infections do not typically transmit from patient to patient or even from a patient back into an environmental

reservoir [8]. This means that multidrug resistance has not typically accumulated over time the way it does in many other pathogens. However, agricultural drug exposure may select for resistance within the environmental reservoir. For example, agricultural exposure to azoles may be contributing to the increasingly common azole resistance in the human fungal pathogen *Aspergillus fumigatus* [9,10]. It remains to be seen whether *C. neoformans* will similarly acquire azole resistance, but a strong possibility exists.

Collectively, we have an urgent need to develop new and synergistic therapies to treat fungal diseases [11]. In *C. neoformans*, this effort has been hampered both by a lack of available experimental tools and by a poor understanding of the essential gene set. Essential genes, particularly those not conserved in humans, are the ideal set of genes for targeted design of novel antifungals. Past systematic efforts to define essential genes in *C. neoformans* by taking advantage of diploid genetics have scaled poorly, resulting in relatively small numbers of defined essential genes [12] (42 of the 6,975 genes). In contrast, most inferences of essentiality were drawn from the ability/inability to obtain gene deletions during the construction of two *C. neoformans* systematic deletion collections [13,14]. However, absence from a deletion collection can be explained by transformation failure rather than essentiality. In addition, many large-scale deletion collections are prone to errors, even in organisms with large communities iteratively fixing them, whether through background mutations [15] or suppressor mutations, such as aneuploidy [16,17]. One study using the *C. neoformans* deletion collection found 14 of 82 tested strains that were hits in a separate CRISPR-based screen failed to produce the correct PCR diagnostic digest pattern that would indicate proper deletion of the relevant gene [18]. Using these data to predict essentiality across the genome is thus likely to be error prone.

One alternate approach to determine essentiality is an approach called transposon mutagenesis sequencing or TN-seq. TN-seq was originally developed in bacteria [19–22] and more recently has been applied to fungi, including *S. pombe* [17,23–25], *S. cerevisiae* [26–29], *Nakaseomyces glabratus* [30], *Yarrowia lipolytica* [31], and *C. albicans* [32,33] among others. This approach inserts one transposon randomly into the genome in each cell within a pool. TN-seq works in haploids, where insertions into essential genes kill the cells, which are thus removed from the pool. Insertions into non-essential genes are tolerated and can be identified using a high throughput sequencing approach (Fig 1A). In pools with millions of cells and in organisms with relatively small genomes, TN-seq can generate saturated maps of genome function. TN-seq has been used broadly to identify essential genes, with some studies using machine learning to predict essentiality of the entire gene set based on an initial subset of confidently assigned genes [32,34].

Further, once a TN-seq insert library has been generated, that library can be selected and resequenced to quantitatively measure the contribution of the remaining nonessential genes to a given phenotype. In eukaryotes, this approach has powered analysis of contributions to heterochromatin formation [23], sexual reproduction [25], general fitness [24], and chemical genetics [26,30]. Similar approaches have been explored using CRISPR in *C. neoformans*, but thus far limitations in effective guide RNAs and resulting low insert densities have limited the statistical power in comparison to TN-seq [18].

Here we have developed a TN-seq system for *C. neoformans* using a modified Ds (dissociation) transposon mobilized by the Ac (activator) transposase from maize. This transposon acts using a cut and paste mechanism and was originally identified by Barbara McClintock [35]. The Ds transposon has been adapted for multiple fungal TN-seq systems [28,32]. We generated a dense insertion library and applied a machine learning model to predict essentiality across the genome. Further, we have selected this library using fluconazole to identify modifiers of drug susceptibility. We demonstrate that inserts into regulatory regions of predicted essential genes can sensitize a strain to drug treatment without altering wild-type fitness. To our knowledge, this is the first use of TN-seq to query essential gene function via perturbation of 5′ regulatory regions in fungi. In sum, we demonstrate that TN-seq enables both identification and analysis of essential genes for their role in drug susceptibility in *Cryptococcus neoformans*.

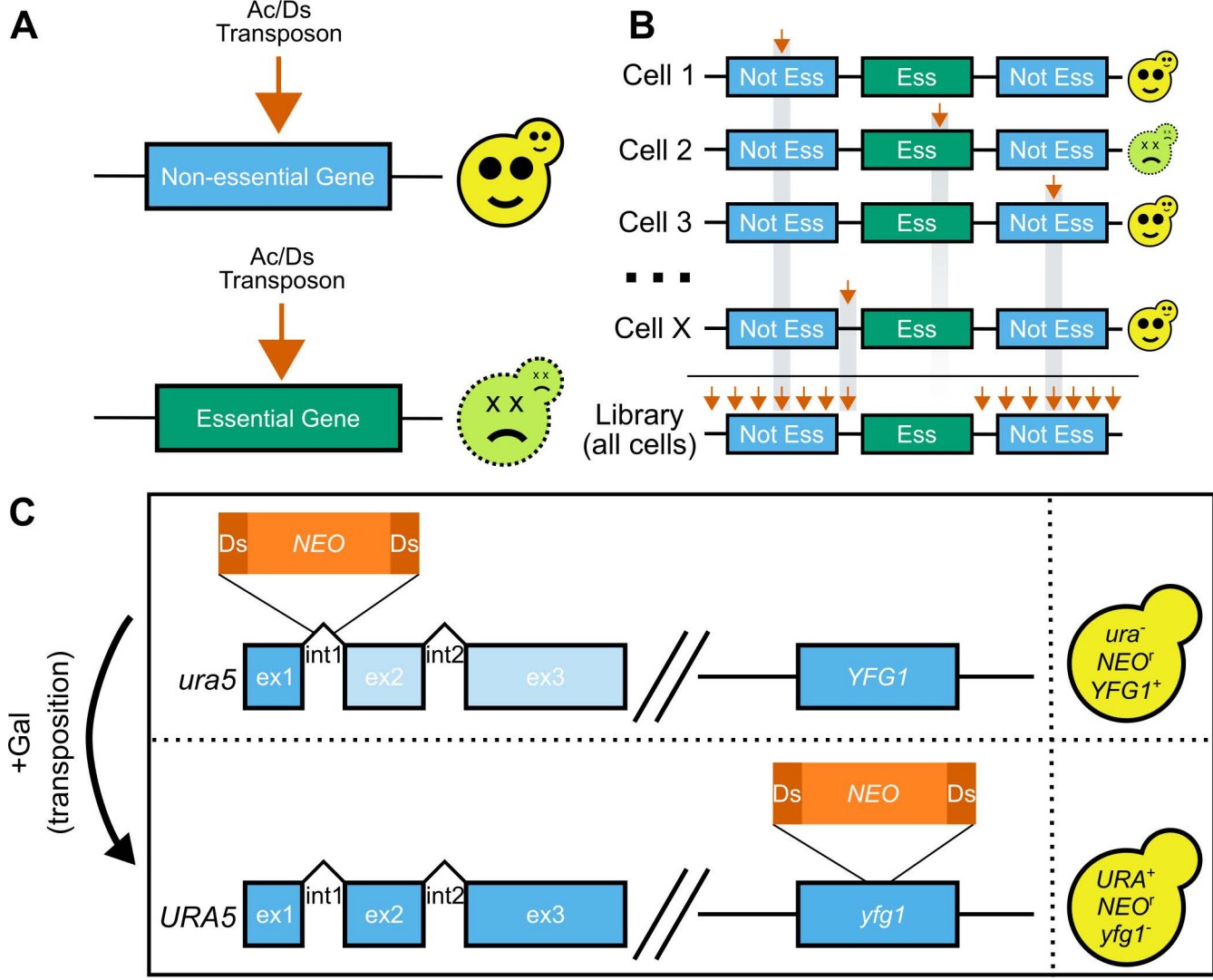

**Fig 1. TN-seq in *C. neoformans*.** (A) Transposon insertions (orange arrow) into nonessential genes results in viable cells. In contrast, insertions into essential genes will result in dead and nonrecoverable cells. (B) TN-seq works by generating a library of cells where each cell has a single independent transposon insertion in a random location. As in **A**, those insertions into essential regions cause the cells to die and are nonrecoverable. As a result, the total library (bottom) is depleted in insertions in essential regions. (C) The Ac/Ds transposon was split into an Ac transposase and a Ds transposon containing a neomycin resistance marker. This Ds transposon was integrated into an intron of *URA5* and the Ac transposase was integrated into the safe haven locus. The resulting stain is *ura−* and neomycin resistant. Upon initiating transposition via growth on galactose, the strain becomes *URA+* and mutant at another locus (depicted here as *YFG1*).

## Results

To determine the set of essential genes in *Cryptococcus neoformans*, we developed a TN-seq assay to generate random insertional mutations across a pool of cells (Fig 1A–1B). Because many of the existing fungal TN-seq systems utilized plasmid systems that were unavailable in *C. neoformans*, we based our approach on a previous TN-seq method in *Candida albicans* [32], with some modifications. In our previous work with a Hermes transposon-based TN-seq system in *Schizosaccharomyces pombe* [25], we observed that transposon insertions (approximately 2.6 kb) were not tolerated within the introns of essential genes. The largest described intron in *S. pombe* is only 819 nucleotides [36], suggesting

that the spliceosome is unlikely to be capable of splicing sequences as large as a transposon insertion. In fact, extension of a functional artificial intron from 36 bases to larger sizes, including both 350 and 252 base introns, showed that increased length introns often fail to splice at all in *S. pombe* [37].

Introns are even slightly smaller in *C. neoformans*, with a mean length of 65 bases [38]. There is also a bias towards RNAi silencing of genes with longer introns [39]. With the assumption that a transposon inserted into an intron would disrupt gene function in *C. neoformans*, we integrated a Ds transposon carrying a neomycin resistance cassette into the first intron of the *URA5* gene in the H99 *C. neoformans* type strain. This insertion disrupts the gene, but gene function is generally restored when the transposon is excised during transposition, even if excision is imperfect (i.e., reading frame does not need to be maintained during repair) (Fig 1C). These features allow us to select both against transposon movement (on media containing 5-FOA) and for transposon excision (on media lacking uracil). The ability to select against transposon movement enabled us to grow large cultures prior to triggering transposition to avoid jackpot events (i.e., clonal expansion of a limited number of independent mutations).

To mobilize the Ds transposon, we integrated a codon-optimized version (designed using Optimizer [40]) of the Ac transposase under the control of the *GAL7* promoter at the neutral safe haven locus [41]. We generated a transposon insertion library by selecting excision events on media containing galactose but lacking uracil so that the transposase could be induced but cells could only grow if they regained the ability to produce their own uracil after transposon excision (Fig 1C, see Materials and methods). We also selected for transposition events via growth in media containing G418 to ensure transposons had reintegrated elsewhere in the genome after excision.

To identify and quantify transposon insertion sites, we amplified the boundaries between transposon insertions and the genome using PCR and sequenced them using Illumina sequencing (modified from [25], see Materials and methods). We integrated random barcodes during the sequencing library preparation, enabling us to eliminate PCR duplicates and more accurately measure transposon insertion frequency within a pool (as in [24]). We mapped only reads containing PCR-amplified Ds fragments to the *C. neoformans* genome. We identified 18,417,912 reads mapping to 1,750,772 unique sites in the genome including coding sequences of 6,190 of the 6,975 total annotated coding genes or 92.5 unique insert sites per kb, on average.

We observed some insertional bias in transposon location. Insertions near the original Ds site in *URA5* were much more common than those in distant locations, suggesting that the transposon was more likely to insert into locations spatially near the original excision site (S1A Fig). This phenomenon called "local hopping" is a common transposon behavior [42] and has been previously observed in multiple transposon mutagenesis systems, including the SATAY TN-seq system of *S. cerevisiae*, which also uses an Ac/Ds transposon system [26]. We also observed some preference for insertion within the upstream regions of genes (S1B Fig). These biases were much stronger in genes we predicted to be essential (see below) than in those we predicted to be nonessential, suggesting that at least a portion of the bias resulted from selection rather than bias in the location of insertion (S1B Fig). Despite these biases, even after discounting inserts on the chromosome containing *URA5* (Chromosome 8), we retained 1,322,416 unique inserts or 75.5 inserts per kb on average. Of these remaining inserts, 37% were genic (40 inserts per kb) and 63% were intergenic (159 inserts per kb). Because the mean gene length in this region (including introns) was 1,906 bases, on average we would still expect 76 insert sites per typical gene, giving us confidence that we could score gene function despite these biases. In contrast, the average gene on Chromosome 8 contains 432 insert sites and intergenic sequences on Chromosome 8 have 472.7 inserts per kb on average.

## Machine learning helps predict essentiality

TN-seq is commonly used in bacteria and more recently in eukaryotes to identify essential genes [43,44]. Approaches can rely on identifying regions with fewer transposon insertions than expected, such that low density regions are scored essential and high-density regions are scored nonessential (reviewed in [44]). Alternately, machine learning approaches

"learn" what an essential and nonessential locus should look like from the data and then score the remainder of the genome. In principle, machine learning methods should be more portable from dataset to dataset and are generally easier to implement [32]. This approach generally relies on using a set of *bona fide* essential genes, which should have few transposon inserts, and *bona fide* non-essential genes, which should have many inserts. In *C. neoformans*, there is a limited set of *bona fide* essential genes. *ERG11* is one such essential gene and the target of the antifungal drug fluconazole. We could very clearly see that Ds insertions within the coding sequence of *ERG11* were not tolerated in our mutant library (Fig 2A). Similarly, transposon insertions were enriched in genes deleted within the *C. neoformans* deletion collection compared with those not contained within the collection ($p = 3.47 \times 10^{-15}$, S1C Fig), with some clear exceptions.

To assess essentiality across the entire *C. neoformans* genome, we built a random forest machine learning model (Fig 2B). We started by taking the set of genes and essentiality predictions reported in Segal and colleagues [32] which contained ortholog and essentiality predictions for *C. albicans*, *S. pombe*, and *S. cerevisiae* genes. We then used FungiDB to identify *C. neoformans* orthologs of these genes [45]. Selecting only genes with identifiable orthologs in all four species left us with 1,653 conserved genes. There are existing essentiality predictions for all species except *C. neoformans* [32]. We reasoned that the majority of these genes conserved over the approximately 600 million years of evolution separating basidiomycetes from ascomycetes would likely have conserved essentiality as well, enabling us to use this set of genes to train our model. Specifically, genes essential or nonessential in *S. cerevisiae*, *S. pombe*, and *C. albicans* were similarly assumed to be essential or nonessential in *C. neoformans*.

We built our model by first parameterizing our TN-seq data set based on those used in previous studies (Fig 2C; [32,34]). We captured 11 variables for each gene, 9 of which described the transposon insertions (total insert frequency within a gene, total insert frequency within a gene normalized to gene length, number of unique transposon insert locations within a gene, number of unique transposon insertion locations within a gene normalized to gene length, total insert frequency within the middle 80% of a gene, total insert frequency within the middle 80% of a gene normalized to gene length, length of largest insert-free gap, length of largest insert free gap normalized to gene length, and total transposon frequency in the 100 kb surrounding the gene) and 2 of which described the gene absent TN-seq information (chromosome and length of gene). We then randomly split our set of 1,653 conserved genes into a training set of 1,322 genes and a validation set of 331 genes. This set includes 600 genes we assumed to be essential and 1,053 we assumed to be nonessential. We randomly chose the training and validation subsets 100 times and built a random forest model from the training subset each time, testing the model against the validation set, and predicting essentiality (Essentiality Scores ranging from 0 to 1, with 1 being essential and 0 being nonessential) for every gene on every iteration. After an initial run (detailed below in Materials and methods) we removed 89 conserved genes from this dataset to remove genes we believed were driving overfitting of the model.

We ran our training and validation 100 times and determined mean precision and recall values from this data (Fig 2D). We then used this data to predict the essentiality score for each gene from the 100 models and used the variation across these models to test whether the mean essentiality score (ES) was statistically different from the 0.5 cutoff. As a result, we define three classes of genes: essential genes (ES > 0.5 and $p < 0.05$ after Bonferroni correction), nonessential genes (ES < 0.5 and $p < 0.05$ after Bonferroni correction) and unclassified genes ($p > 0.05$). Our data achieved a mean precision (true positives/(true positives + false positives)) of 0.94 and mean recall (true positives/(true positives + false negatives)) of 0.87. This model compares favorably to one used for *C. albicans* (Precision 0.71, Recall = 0.62) [34]. Given that some differences are likely biological (i.e., a conserved essential gene is not essential in *C. neoformans*) rather than model defects, these scores are quite strong. Notably, this revised model was capable of assigning essentiality to many genes that differed from that assumed in the initial conservation-based assumptions (Fig 3B, 3C, discussed below).

We also extracted the importance of each value to the model's predictions (Fig 2E). Our model was most strongly impacted by length adjusted metrics, including the strongest metric, which was the percentage of the gene that was covered by the largest insert free gap. Interestingly, the frequency of transposon insertions in the local environment

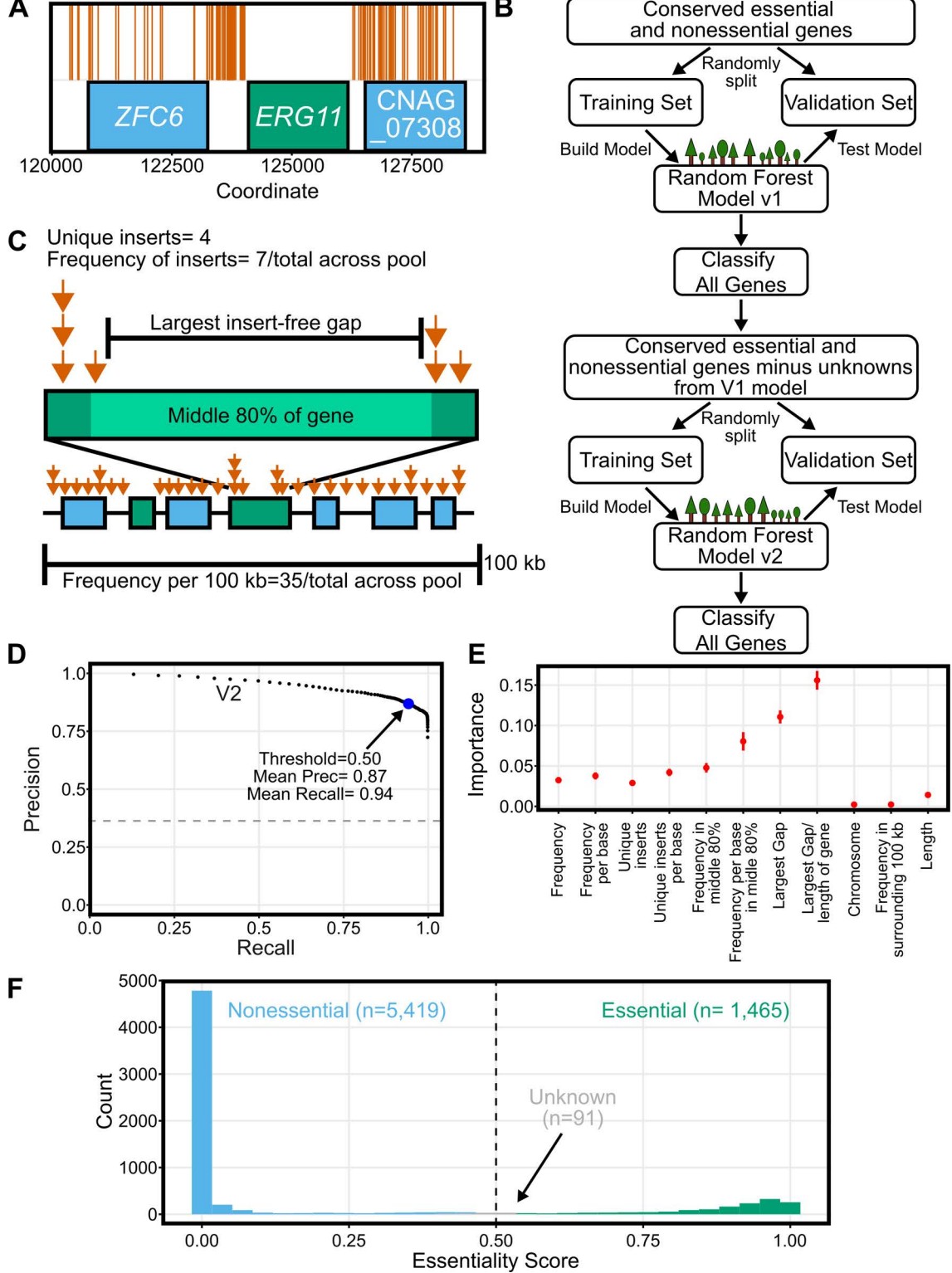

**Fig 2. TN-seq enabled prediction of gene essentiality.** (**A**) 176 unique transposon insertions (orange vertical lines) are plotted along a region of Chromosome 1 centered on the known essential gene *ERG11* and showing two flanking predicted nonessential genes. For nine of the displayed sites, we recovered transposon insertions in both orientations. (**B**) Flow chart depicting the random forest approach to classifying gene essentiality. (**C**) Schematic

illustrating parameters that describe each gene within the TN-seq data for machine learning. **(D)** Precision recall curve describing tradeoff between precision and recall for the random forest model. Each point is the mean of 100 replicates where the training data was randomly split into training and validation sets. The threshold was then varied by 0.01 from 0.01 to 0.99 for each set. **(E)** The importance of each feature for the V2 model is plotted. Each importance value is also calculated from the same 100 replicates of the training data. Error bars indicate standard deviation. **(F)** Histogram of the essentiality prediction score for the entire gene set of *C. neoformans* based on the mean of 100 replicates. Data underlying **A** can be found in S1 Data at 10.5281/zenodo.15264486. Data underlying **F** can be found in S1 Table. Raw data underlying **D–E** can be found in corresponding excel sheets at 10.5281/zenodo.15264486.

(surrounding 100 kb) and the specific chromosome had very little impact on the model, despite the biases in transposon insertion frequencies across the genome that we observed.

We then used our revised model to predict essentiality scores for every *C. neoformans* gene (Fig 2F, S1 Table). Using the statistical approach described above, we predicted 1,465 essential genes and 5,419 nonessential genes, with 91 unclassified genes. There was reasonably good overlap between our approach and the existing deletion collection. 76.4% ($n$ = 4,139) of the genes we scored as nonessential are deleted in the collection and 92.6% ($n$ = 1,357) of the genes we scored as essential are not in the collection [14]. Still, there are numerous discrepancies. 23.6% of the genes ($n$ = 1,280) we predict are non-essential are missing from the deletion collection. This is not inherently surprising as there are known nonessential genes missing from the whole genome deletion collection (e.g., *BAP1* and *PAN1*, TFs [46]; *PBS2* and *MEC1*, kinases [47]; *PTC5* and *HPP2*, phosphatases [48]; and *LMP1* [49]). Similarly, no claims have been made that failure to construct a deletion distinguishes essentiality rather than experimental failure. In fact, the builders of the deletion collection instead suggest missing nonessential genes may be enriched for those with significant growth defects [14].

More surprising is that we classified 108 genes as essential that were present in the deletion collection. We hypothesized three potential explanations for this discrepancy. First, the construction of our mutant library required growth on SC-URA + GAL media, which contains no uracil and galactose as a primary carbon source. Any genes required for growth on this medium would score as essential in our assay. For example, some genes required for galactose metabolism, like *GAL10* (CNAG_06050) scored as essential in our model despite being present in the deletion collection. Similarly, our model failed to classify *GAL7* (CNAG_06052) as essential or nonessential, instead landing in our unclassified category. Additionally, our assay is performed in a pool. Insertion mutants with severe growth defects may be outcompeted by other strains and not be represented in the final TN-seq library, thus scoring as essential. Finally, some strains in the deletion collection may not have the intended gene deleted.

We tested putative mutants from the deletion collection for five of the genes we classified as essential with a range of essentiality prediction scores (ranging from 0.58 to 0.98) to determine which of the models above may explain the discrepancy. We started by testing all five mutants, using 5′ and 3′ junction PCRs and in gene PCRs (S2A Fig). For two of the mutants (CNAG_06887 (essentiality score (ES) = 0.58) and CNAG_05292 (ES = 0.95)), we were unable to PCR validate any of 5 independent colonies from the deletion collection, including positive PCR products for an in-gene PCR, suggesting that these strains did not contain deletions of the indicated gene (S2B, 2C Fig). For the other three genes (CNAG_02190 (ES = 0.76), CNAG_04763 (ES = 0.91), CNAG_00996 (ES = 0.98)) we validated their deletion in the deletion collection, including the absence of an in-gene PCR product, indicating our TN-seq approach misclassified the genes (S2D Fig).

We tested the growth of the deletion alleles of the three misclassified genes. We found that the *cnag_02190Δ* deletion had a growth defect that was exacerbated on YP + GAL and SC-URA + GAL (S3A Fig). The *cnag_04763Δ* deletion had a very mild growth defect on YP + GAL that was exacerbated on SC-URA + GAL and SC-URA, while the *cnag_00996Δ* deletion made normal size colonies on both media but with unusual morphology on SC-URA + GAL (S3C Fig). To further query the growth of these mutants we used competition assays where we competed the NAT resistant mutants against an unmarked KN99 wildtype strain in SC-URA + GAL media, similar to the media used in the original TN-seq assay, except in liquid. Both *cnag_02190Δ* and *cnag_04763Δ* were outcompeted by the wildtype while *cnag_00996Δ* grew at relatively the

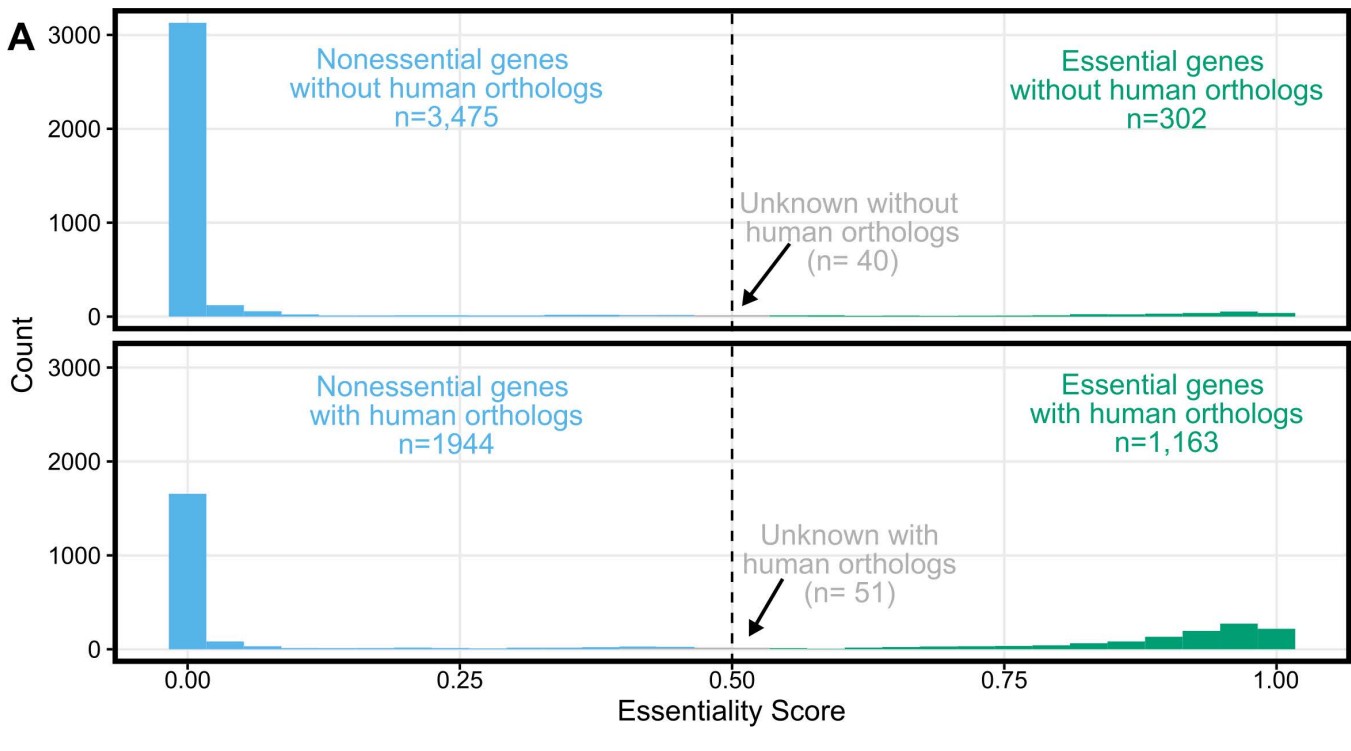

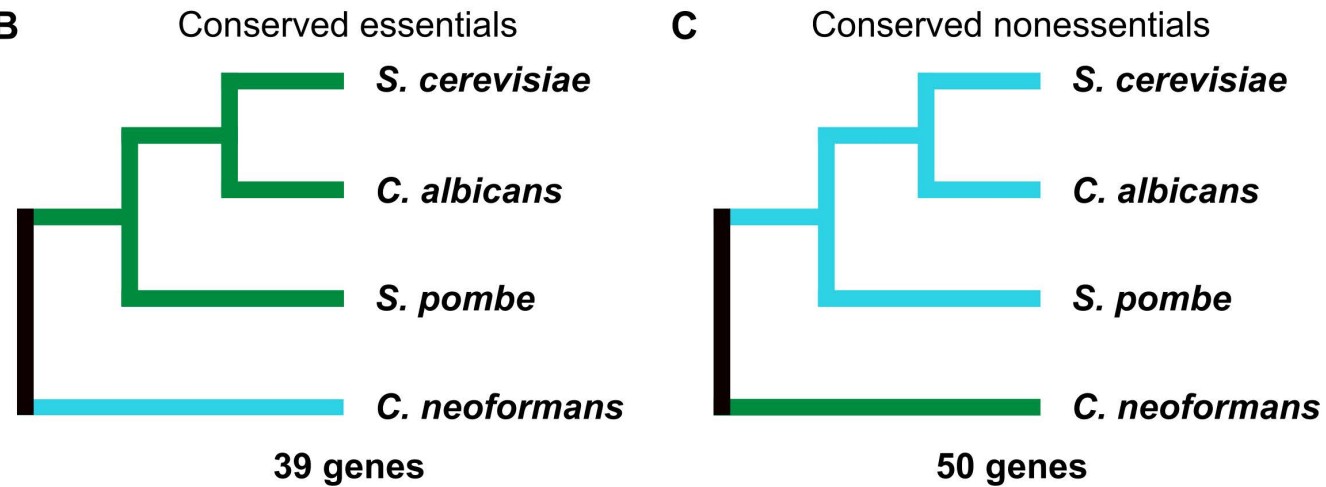

**Fig 3. Evolution and conservation of essentiality. (A)** Histogram of essentiality score as in Fig 2F, split by conservation status in humans. Genes without human orthologs are shown at top and those with human orthologs are shown below. **(B)** Genes with orthologs in *S. cerevisiae*, *C. albicans*, *S. pombe* and *C. neoformans* where the orthologs are essential in all three ascomycetes but predicted dispensable in *C. neoformans*. **(C)** Genes with orthologs in *S. cerevisiae*, *C. albicans*, *S. pombe* and *C. neoformans* where the orthologs are dispensable in all three ascomycetes but predicted essential in *C. neoformans*. Trees depicted in **B–C** are not inferred here but are based on the accepted relationship between these fungal groups [97]. Data underlying **A** can be found in S1 Table. Data underlying **B–C** can be found in S4 Table.

same rate as the wildtype (S3B Fig). Our data suggest that the discrepancies are likely a mix of at least two of our three models, although we lack a satisfying explanation for why CNAG_00996 scored as confidently essential in our hands (essentiality score of 0.867) but was clearly not essential and lacked a growth defect in our relevant media conditions.

Finally, Ianiri and colleagues characterized essential *C. neoformans* genes in their 2015 study. They identified 21 genes that were described as essential in previous publications and additionally identified 21 more essential genes by looking for the failure to recover mutant haploid progeny from heterozygous mutants [12]. Our analyses predicted 39 of those 42 genes to be essential as well. One possible explanation for the three exceptions is that the assay employed by Ianiri and colleagues required mutant spores to germinate. Mutants defective in germination would score as essential in their assay but not ours. Alternately, in our pooled assay, essential gene mutants could be complemented by neighboring wildtype cells through production of diffusible factors. This type of complementation would not occur in the assay used by Ianiri and colleagues. None of the three genes we predict as nonessential are found in the deletion collection, perhaps favoring an issue with the TN-seq assay rather than a germination defect.

## Conservation of essential genes

One of the primary goals of identifying essential genes in fungal pathogens is to identify potential targets for antifungal drug development. Because fungi are so closely related to humans, many genes are present in both pathogens and their hosts, making these genes poor choices for antifungal development. We used FungiDB [50] to annotate if each gene in the *C. neoformans* genome had a human ortholog. This approach identified 3,817 genes without human orthologs and 3,158 genes with human orthologs (Fig 3A). However, we identified only 302 genes that we scored as essential that also lacked human orthologs, compared to the 1,163 predicted essential genes with human orthologs (Fig 3A). The predicted essential genes lacking human orthologs were enriched for GO annotated functions in various biosynthetic processes (such as "small molecular biosynthetic process", *p* = 1.87*e*−21; "organic substance biosynthetic process", *p* = 4.53*e*−10; "biosynthetic process", *p* = 8.67*e*−10) and also cell division (such as "cell cycle", *p* = 3.53*e*−5; "cell division", *p* = 7.83*e*−5; chromosome segregation, *p* = 7.23*e*−4) (S2 Table).

Previous work identified 694 genes whose orthologs were essential in ascomycete yeasts *S. cerevisiae*, *S. pombe* and *C. albicans* [32]. We used FungiDB [50] to identify orthologs for these genes and found that 600 of these genes also had orthologs in *C. neoformans,* which is a basidiomycete. Our model predicted that 558 of these genes were also essential in *C. neoformans*. Once again, we used FungiDB [50] to identify human orthologs and only 34 of these 558 conserved essential genes lacked human orthologs. This compares with 30 genes identified by Segal and colleagues for a similar comparison (essential in *S. cerevisiae*, *S. pombe*, *C. albicans*, ortholog in *C. neoformans*, no ortholog in humans) [32]. Segal and colleagues used a different, relatively stringent method to identify orthologs and it is likely not surprising that they identified fewer genes, even without considering essentiality in *C. neoformans*. In fact, 17 of the genes identified here are predicted by the *Saccharomyces* Genome Database (SGD [51]) to have human orthologs (S3 Table). The genes we identified as conserved essential genes lacking human orthologs include genes involved in inositol phosphoceramide metabolism, including an ortholog of *AUR1*, the target of the antifungal drug aureobasidin A [52]. Further exploration of these genes, as well as those specific to *C. neoformans*, will likely be fruitful in targeted design of broad or narrow spectrum antifungal agents, respectively. A number of genes have roles in chromosome segregation or core metabolism and could make highly effective drug targets.

When we compared our predicted essential genes in *C. neoformans* to those of the other fungi above, we identified 39 genes that had orthologs that scored as essential in the other three fungi, but our model scores as nonessential (Fig 3B, S4 Table). Twenty-eight of these 39 (71.8%) genes are represented in the deletion collection, which is similar to the average representation of deletions in the nonessential genome (76.4% of the scored nonessential genes have a deletion in the collection; *p* = 0.570, Fisher's exact test). The differential essentiality of a number of these orthologs may be explained by paralogy (gene duplicates). For example, *C. neoformans* CNAG_01675 is confidently nonessential, unlike the identified orthologs in the other species. But whereas the gene is single copy in the other three fungi, there are three paralogs in *C. neoformans*. Two of these paralogs, CNAG_01675 (ES = 0.000003, *p* = 0) and CNAG_02590 (ES = 0, *p* = 0), are in our set of predicted nonessential genes while the third paralog CNAG_06770 is confidently predicted to be essential

(ES = 0.987, $p = 1.57 \times 10^{-168}$). Seventeen of the genes that are predicted to be nonessential in *C. neoformans* but are essential in the other fungi have paralogs in *C. neoformans*. However, the reasons for the non-essentiality specifically in *C. neoformans* for the 22 other genes lacking paralogs are not obvious. For example, the predicted nonessential *C. neoformans* genes CNAG_03878 and CNAG_07951 are orthologs of essential genes involved in 60S ribosomal biogenesis, which should be an essential process. Further exploration of many of these discrepant genes will likely be enlightening.

We also queried the essentiality in *C. neoformans* of 1,053 genes that were classified nonessential in all 3 of the other fungi. Fifty of these genes scored as essential in *C. neoformans* (Fig 3C). These genes are similar in length to the rest of the gene set (average length of 2049 bases to 1903 bases for the entire genome) suggesting that false positives are not arising because of short genes that are missing inserts by chance. However, 11 of the 50 are present in the deletion collection, which is statistically enriched compared to the whole set of our predicted essential genes ($p = 0.0012$, Fisher's exact test), suggesting that some of these discrepancies are likely false positives for essentiality in our predictions. Some of these genes may be false positives resulting from the TN-seq approach. For example, *UGE1* and *UGE2* are both homologs of *GAL10* in *S. cerevisiae,* and presumably score as essential in our assay because they are required for growth on galactose, which was used to induce transposition. Both are present in the deletion collection. However, as above, other examples are less easily explained. For example, CNAG_06737 encodes an ortholog of *VPS16*, which is not essential in any of the three ascomycete yeasts, does not appear to have an annotated role in galactose metabolism, and does not appear to have paralogs in any of the three ascomycete species. CNAG_06737 is also missing from the *C. neoformans* deletion collection. While *vps16+* is required for high temperature growth in *S. pombe* [53], at normal temperatures mutants appear to grow faster than wildtype [54]. Understanding how *VPS16* and other genes have become essential in *C. neoformans* will likely reveal unique elements of adaptation that differentiate basidiomycetes like *C. neoformans* from the ascomycete model yeasts.

## TN-seq identifies genes involved in growth in fluconazole

We also used our TN-seq libraries to assay the sensitivity/tolerance of *C. neoformans* mutants to fluconazole on a genome-wide scale (Fig 4A). Fluconazole is one of the primary drugs used to treat many fungal pathogens. While the frontline therapy for cryptococcosis is a combination therapy of 5FC and amphotericin, azoles are often used instead either alone or in combination with amphotericin. Fluconazole is also typically used for long term suppression or prophylactic treatment. Azole drugs function by inhibiting the Erg11 protein, which both reduces the synthesis of ergosterol and promotes the production of toxic intermediate sterols. We performed growth curves with serial dilutions of fluconazole to determine the half-maximal inhibitory concentration ($IC_{50}$) for our starting strain when grown in YPD (S4 Fig). We then split our TN-seq library to grow in YPD with either 0.1% DMSO or the $IC_{50}$ level of fluconazole (13.5 µg/mL), which is dissolved in DMSO (final concentration of 0.1%). After two rounds of dilution to an OD of 0.1 followed by growth to saturation (approximately 8 doublings per round, 16 total) in either DMSO or fluconazole, we determined the transposon landscapes in both populations. We compared these landscapes to that of the starting library that we used for essential gene prediction above. To control for differential growth rates and the effect of DMSO, we compared the frequency changes at every site in the genome after growth in fluconazole to the frequency changes after growth in 0.1% DMSO. To test for statistical significance, we compared the distribution of frequency changes across a gene to those of the presumably neutral intergenic sites across the entire genome ($n = 454,341$). Because some individual insert sites can be noisy, especially at low insertion frequencies, we only scored genes with at least 5 unique insertion locations. This filtering removed 1,228 of our predicted essential genes but still allowed us to assay 237 predicted essential genes. The filtering also eliminated 387 predicted nonessential genes and 64 "unclassified" genes. In total, we were able to score the effect of fluconazole on mutants of 5,296 genes (S1 Table).

We identified 398 genes that were specifically enriched or depleted for inserts after treatment ($p < 0.05$ after a Bonferroni correction, $n = 5,296$) (Fig 4B). Of these, 185 genes had inserts that on average increased in frequency in

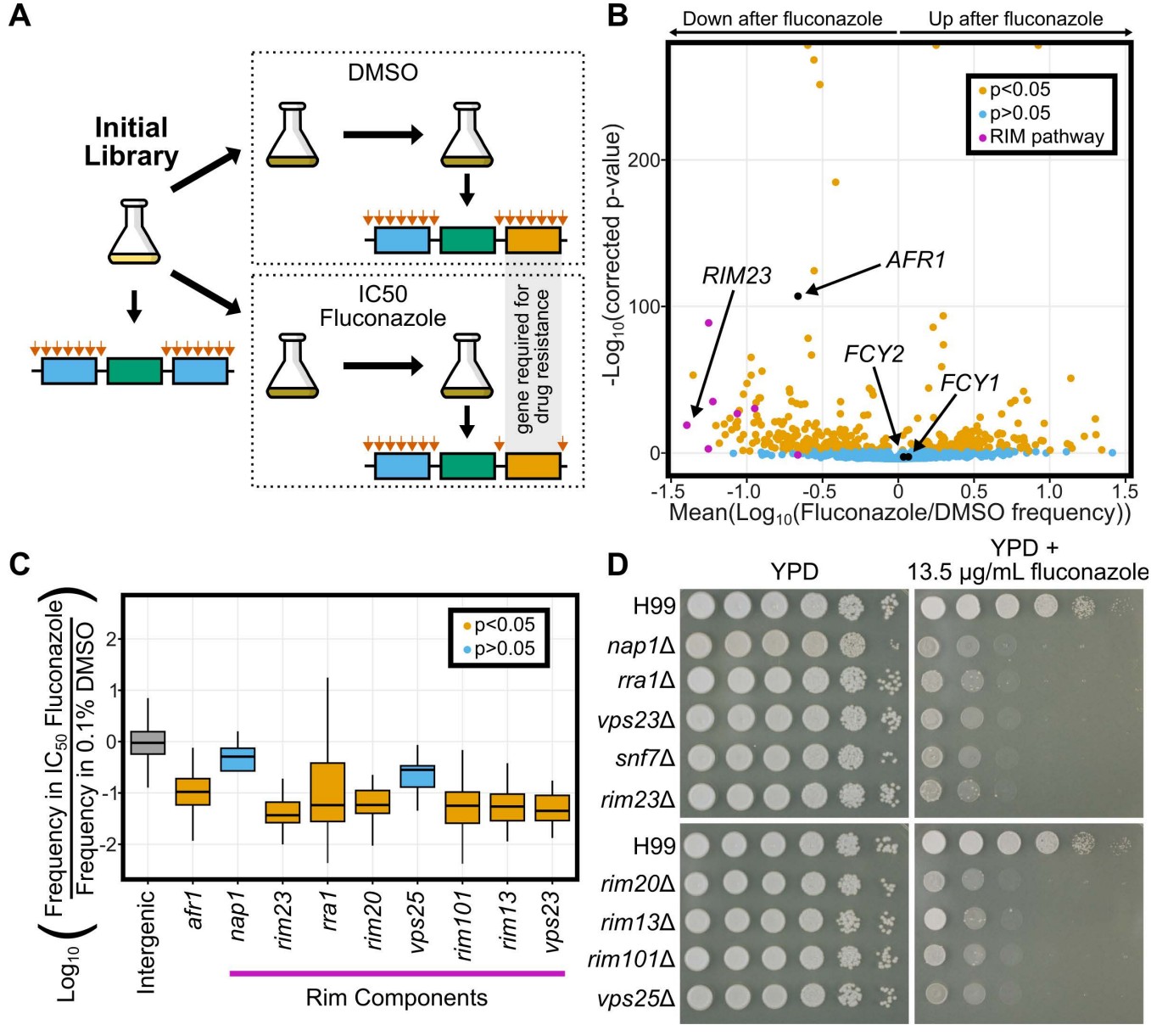

**Fig 4. Using TN-seq to assay genetic contribution to fluconazole resistance. (A)** TN-seq libraries were selected with $IC_{50}$ levels of fluconazole dissolved in DMSO or with just the equivalent amount of DMSO as a control. Libraries were sequenced at time 0 and after two days growth in DMSO or fluconazole to identify genes with differential transposon insertion frequencies after selection. **(B)** Volcano plot of 5296 genes with 5 or more insert sites (out of 6,975 *C. neoformans* genes) displaying the mean $\log_{10}$ (Fluconazole/DMSO) value on the *x*-axis and the $-\log_{10}$ (Bonferroni corrected *p*-value) on the *y*-axis. Individual genes are shaded orange if the distribution of inserts is statistically different from the distribution of inserts into noncoding regions ($p < 0.05$ via Mann–Whitney *U* test after Bonferroni correction). Genes that are not statistically different are shaded blue. **(C)** Boxplot displaying distribution of $\log_{10}$-adjusted fold changes in insert density (i.e., frequency in fluconazole/frequency in DMSO). Boxplots show first quartile, median, third quartile. The whiskers show the range to a maximum of 1.5 times the interquartile range above and below the first and third quartile, respectively. Outlier data points (outside the whiskers) are not displayed. Not displaying outliers results in 15,902 of 454,341 intergenic sites, 1 of 107 sites from *afr1*, 1 of 4 sites from *nap1*, 1 of 34 sites from *rim23*, 2 of 127 sites from *rra1*, 12 of 74 sites from *rim20*, 0 of 5 sites from *vps25*, 5 of 77 sites from *rim101*, 5 of 151 sites from *rim 13*, and 1 of 11 sites from *vps23* not being displayed although those data were considered in the statistical analyses. Snf7 is not shown because there were 0 inserts between the start and stop codons. Inserts in intergenic regions are indicated in grey, genes where inserts were significantly depleted after fluconazole treatment are shown in orange (*afr1, rim23, rra1, rim20, rim101, rim13, vps23*) and genes where inserts were not significantly depleted after fluconazole are shown in blue (*nap1 and vps25*). Notably, both *nap1* and *vps25* had very low numbers of inserts that limited statistical power. **(D)** Spot dilution assays with 5 µL spots plated. The initial leftmost spot is of $OD_{600} = 20$ culture and each successive spot is a 10-fold

dilution, so that the final spot should be $10^5$ less concentrated than the first. Both plates were spotted on the same day with the same dilution series. All plates were imaged after 48 h at 30°C. Mutants were spotted on two separate plates for YPD and fluconazole media, each with a wildtype H99 control present. Data underlying **B** can be found in S1 Table. Data underlying **C** can be found in S1 Data at 10.5281/zenodo.15264486. Original images in panel **D** can be found in the Stowers Original Data Repository at http://www.stowers.org/research/publications/libpb-2480.

fluconazole, meaning the mutants had a relative growth advantage in the presence of the drug, while 213 genes had inserts that on average decreased in frequency, meaning the mutants had enhanced sensitivity to the drug. We recovered known genetic interactions for fluconazole. For example, *AFR1* is well known as a major efflux pump for fluconazole [55]. Transposon insertions in *AFR1* were significantly depleted after fluconazole selection (log(fluconazole/DMSO) = −0.971, $p = 1.52e−57$) (Fig 4B). In contrast, insertions in either of two genes, *FCY1* and *FCY2*, involved in 5FC toxicity but not fluconazole toxicity were unaffected (*FCY1*: log(fluconazole/DMSO) = 0.0474, $p = 0.171$; *FCY2*: log(fluconazole/DMSO) = 0.0276, $p = 0.162$) (Fig 4B) [56,57]. Together, these data gave us confidence to explore the hits from our assay.

Interestingly, among the strongest hits for fluconazole sensitivity were mutants in components of the *RIM101* pathway, which had not previously been implicated in fluconazole sensitivity. In *C. neoformans*, the *RIM101* pathway has a canonical role in pH sensing and is important for capsule production [58]. Of nine described components of the *RIM101* pathway in *C. neoformans* [59], we assayed eight here and six were sensitive to fluconazole (Fig 4C). Some of these *RIM101* pathway genes exhibited similar magnitude insertion losses to the antifungal efflux pump *AFR1* [55]. We had limited statistical power for some of these *RIM101* pathway genes (*SNF7* had 0 coding transposon insertions and is not pictured) because insert sites were relatively low in number even prior to selection with fluconazole. To further investigate these hits, we assayed deletion mutants of nine *RIM101* pathway genes in spot dilution assays. All nine mutants exhibit fluconazole sensitivity compared to wildtype (H99), even those that did not score as statistically significant in our TN-seq assay (Fig 4D).

## Assaying essential genes using TN-seq

Essential genes are not represented in deletion collections, so these collections are not useful for directly screening essential gene functions. TN-seq mutant libraries can similarly not maintain null alleles of genes with essential functions. However, we hypothesized that insertions in regulatory regions of essential genes, which can sometimes be tolerated, may generate sensitized, or hypomorphic alleles. If so, then any selection affecting a pathway involving a given essential gene could favor or disfavor insertions in the regulatory regions of that gene. To test this hypothesis, we examined inserts near *ERG11* in our datasets (Fig 5A)*.* We reasoned that mutants with transposon insertions in the region upstream of *ERG11* should be specifically sensitive to fluconazole and thus should be depleted when a TN-seq library is treated with fluconazole (Fig 5A). This model was supported by our TN-seq data. Mutants with inserts into the 300 base pair region directly upstream of *ERG11* were well tolerated in our control condition (YPD with 0.1% DMSO) but were selected against with the addition of fluconazole ($p = 4.14 \times 10^{-8}$; Fig 5B). In contrast, mutants with inserts into the 300 base pair region downstream of *ERG11* were not depleted by the addition of fluconazole ($p = 0.383$). Insert orientation did not appear to play a substantial role in this phenotype. This suggests that the phenotype is the result of loss of function, rather than insertion of a cryptic promoter carried by the Ds transposon driving gain of function phenotypes as in the SATAY system in *S. cerevisiae*, which also uses the Ds transposon [26]. Aside from the substantial evolutionary difference between these two fungi, our system uses a *C. neoformans* specific resistance marker, as opposed to the *S. cerevisiae* markers which are known to include bidirectional promoters capable of driving neighboring genes [60].

We decided to use the TN-seq data to guide construction of hypomorphic alleles of essential genes, using *ERG11* as a test case. Inserts across the region upstream of *ERG11* allowed us to define the regions where inserts would or would not disrupt *ERG11* function. We generated two targeted mutants, with inserts either 296 bases upstream of the start codon that were predicted not to alter fluconazole resistance or 77 bases upstream which were predicted to decrease fluconazole resistance. Consistent with our predictions, the −77 insertion generated strains which are highly sensitive to

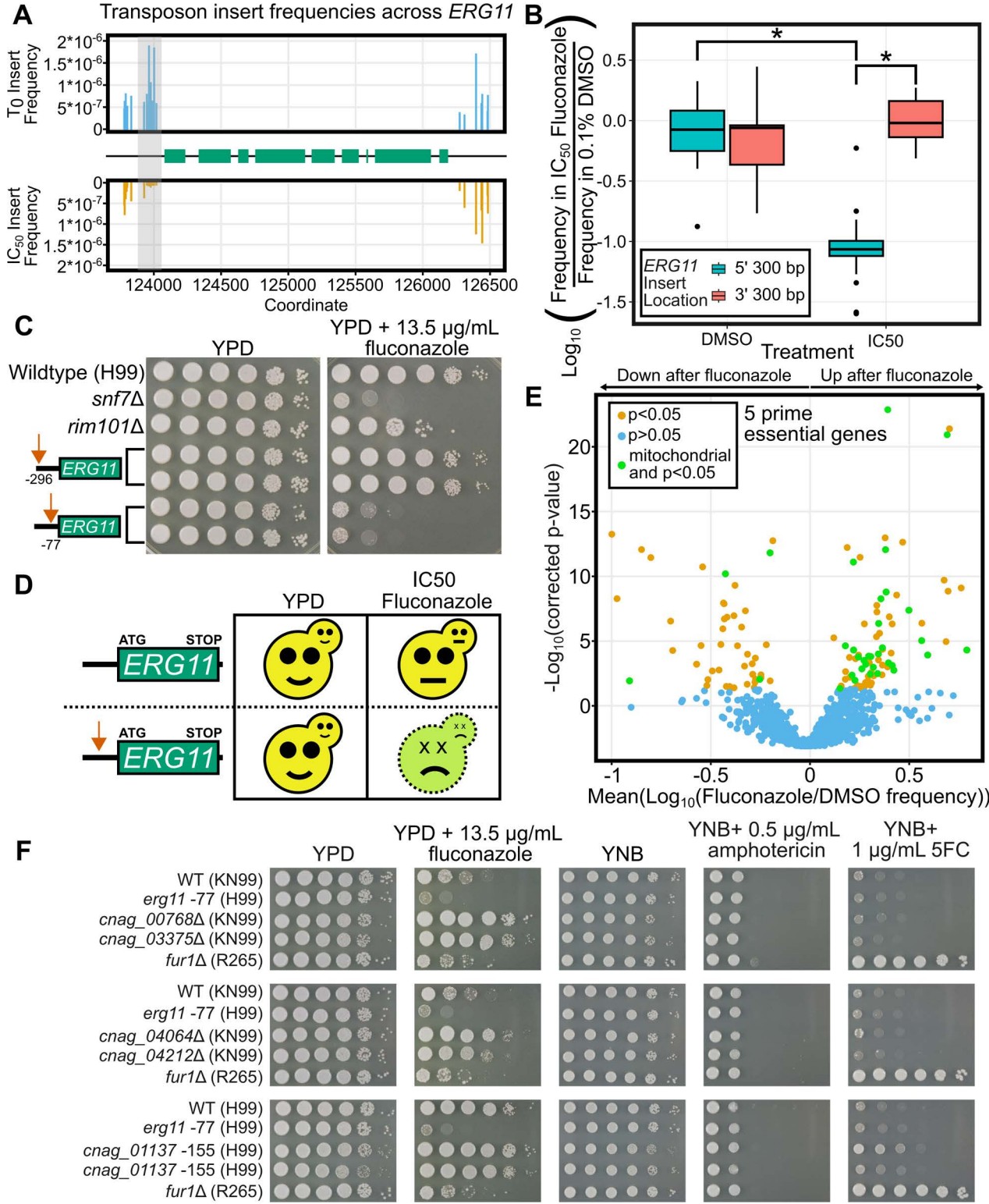

**Fig 5. Regulatory inserts enable assays of essential gene function. (A)** Plot showing transposon insert frequencies in the region surrounding the *ERG11* gene at either time zero or after two passages in IC$_{50}$ levels of fluconazole in YPD. Shaded area shows the upstream region where inserts are less fit in fluconazole. **(B)** Boxplot displaying distribution of log$_{10}$-adjusted fold changes in insert density (i.e., frequency in fluconazole/frequency

in DMSO). Each column shows inserts only within a 300 base pair region either immediately upstream of the start codon or downstream of the stop codon. Boxplots show first quartile, median, third quartile. The whiskers show the range to a maximum of 1.5 times the interquartile range above and below the first and third quartile, respectively. Outliers are displayed as individual datapoints. **(C)** Spot dilution assays with 5 µL spots plated. The initial leftmost spot is of $OD_{600}$ = 20 culture and each successive spot is a 10-fold dilution, so that the final spot should be $10^5$ less concentrated than the first. Both plates were spotted on the same day with the same dilution series. YPD plates were imaged after 48 h at 30°C and fluconazole plates were imaged after 72 h at 30°C. **(D)** Schematic of model for 5′ transposon insertions. Wildtype cells (top row) should grow well on YPD and be moderately impaired (approximately 50%) by an $IC_{50}$ level of fluconazole. Cells with a transposon insertion in the 5′ regulatory region of *ERG11* should grow well on YPD but be highly sensitive to $IC_{50}$ levels of fluconazole. **(E)** Volcano plot of 1,251 predicted essential genes with 5 or more insert sites in the 300 base pairs upstream of the start codon (out of 6,975 *C. neoformans* genes) displaying the mean $\log_{10}$ (Fluconazole/DMSO) value on the *x*-axis and the $-\log_{10}$ (Bonferroni corrected *p*-value) on the *y*-axis. Individual genes are shaded orange if the distribution of inserts is statistically different from the distribution of inserts into noncoding regions genome-wide ($p < 0.05$ via Mann–Whitney *U* test after Bonferroni correction). Genes that are not statistically different are shaded blue. **(F)** Spot dilution assays with 5 µL spots plated. The initial leftmost spot is of $OD_{600}$ = 20 culture and each successive spot is a 10-fold dilution, so that the final spot should be $10^5$ less concentrated than the first. All plates were spotted on the same day with the same dilution series. YPD and YNB plates were imaged after 48 h at 30°C and drug plates were imaged after 72 h at 30°C. Data underlying **A** and **B** can be found in S1 Data at 10.5281/zenodo.15264486. Data underlying **E** can be found in S1 Table. Original images in panels **C** and **F** can be found in the Stowers Original Data Repository at http://www.stowers.org/research/publications/libpb-2480.

fluconazole while growing at wildtype levels on rich media (Fig 5C), while the −296 insertions generated strains that grew at wildtype levels on fluconazole and rich media (Fig 5D).

We then extended these analyses to assay regulatory inserts on a genome-wide scale, similar to examining coding sequence insertions. We were able to examine 5′ regulatory insertions for 6194 genes (compared with 5,296 for coding insertions). 1,251 genes we predicted were essential had at least 5 unique insertion sites within the 300 bp upstream of the start codon compared with only 237 predicted essential genes that had at least 5 unique insertion sites within the far larger coding sequences. Of those 1,251 predicted essential genes, 126 had inserts significantly enriched (*n* = 79) or significantly depleted (*n* = 47) within 300 base pairs upstream of the start codon after fluconazole treatment (*p* < 0.05 after Bonferroni correction, *n* = 1,251).

Interestingly, when we examined predicted essential genes with significant increases in insert frequency at the 5′ end in response to fluconazole (i.e., mutants were more resistant), we saw a number of genes with mitochondrial functions (30 of 79 genes, Fig 5E). These genes encoded components of the electron transport chain, ATP synthase, mitochondrial ribosomal subunits, and others. In contrast, predicted essential genes with significant decreases in insert frequency at the 5′ end had significantly fewer genes with mitochondrial function (4 of 47 genes; *p* = 0.0003, Fisher's exact test). There are no obvious differences between the 30 genes we identified where inserts in the 5′ end conferred resistance and the four genes where inserts in the 5′ end conferred sensitivity (a mitochondrial ribosomal subunit, a malate dehydrogenase, a phosphoglycerate kinase, and an inositol phosphorylsphingolipid-phospholipase C), but there are reasonable models that could explain this pattern. For example, genes could fall into this category if they play a role in both mitochondrial function and cell wall integrity. Together, these results suggest that loss of mitochondrial function generally makes cells more resistant to fluconazole. This is consistent with past work showing that rotenone, a specific inhibitor of the electron transport chain complex I, is strongly antagonistic with fluconazole in *C. neoformans* [61]. Our genetic data suggests this phenotype is broader than just the electron transport chain and includes general mitochondrial function.

In other organisms, general mitochondrial disfunction can induce general stress responses that increase drug efflux and thus result in multidrug resistance. Point mutants in *COX10* cause resistance to azoles, terbinafine and caspofungin in *A. fumigatus* [62]. In *S. cerevisiae*, mitochondrial disfunction induces expression of pleiotropic drug efflux pumps [63]. To test whether the phenotypes we observed were specific to fluconazole or generalizable to many antifungal agents, we tested deletions of four nonessential mitochondrial genes (*CNAG_00768* (unnamed, ETC complex I associated protein), *CNAG_03375* (*LSC1*), *CNAG_04064* (*MIX17*), *CNAG_04212* (*MIC19*)) from the Madhani collection [14] and generated two new 5′ insertion alleles of an essential mitochondrial gene (CNAG_01137, *ACO1*). All five displayed resistance to fluconazole compared to their parental strains (Fig 5F), although the phenotype of the CNAG_01137 −155 alleles was

clearer at higher doses of fluconazole (S6 Fig). However, aside from a subtle phenotype on 5FC for one of the two inde-pendent CNAG_01137 −155 alleles, none of the mutants tested showed increased resistance to 5FC or amphotericin, the two other commonly used antifungals for treating Cryptococcosis (Fig 5F). We thus favor the interpretation that this phenotype is specific to fluconazole and unlikely to be the result of a general increase in drug efflux. In fact, we observed multiple mutations in the TCA cycle that would predict a buildup of acetyl-CoA. Acetyl-CoA can either feed into the TCA cycle or be used to synthesize ergosterol. Slowing or stopping mitochondrial metabolism may result in increased usage of acetyl-CoA by ergosterol biosynthesis, helping to compensate for the reduced activity of *ERG11* during fluconazole treatment.

### Shiny app allows navigation of fluconazole TN-seq data

As a resource for the *C. neoformans* community, we developed an interactive Shiny app to help readers navigate our TN-seq data (S7 Fig, https://simrcompbio.shinyapps.io/Crypto_TN_seq_viewer/). This app includes data for all genes, however, genes that lack sufficient inserts (and thus did not pass our depth cutoffs) will display only a subset of the plots. The plots enable visualization of both the distribution and frequency of transposon insertions within a gene as well as how they respond to fluconazole selection. This should enable the *C. neoformans* community to more easily navigate this dataset.

## Discussion

Here we present a TN-seq system for *Cryptococcus neoformans* that enables genome-wide assays of gene function. We used our TN-seq data to predict gene essentiality and to select and identify genes contributing to fluconazole sensitivity and resistance. We identified numerous essential genes as well as novel modifiers of fluconazole tolerance and resis-tance. Finally, we showed that transposon insertions can help guide the construction of hypomorphic alleles for essential genes and allow analysis of essential genes in a screen-like fashion.

### Essential gene identification

Identifying essential genes is important because these genes can serve as targets for antifungal drug discovery. We pre-dicted 1,465 essential genes using a random forest machine learning model trained on our TN-seq data. This represents 21.0% of the genes tested. Approximately 18% of the *S. cerevisiae* genome is essential [64] and 26% of the *S. pombe* genome is essential [65]. There are multiple factors that could contribute to these differences in essentiality. *S. pombe* has far fewer genes (n = 5,134)[66] than *S. cerevisiae* (n = 6,275)[67] and *C. neoformans* (n = 6,975)[45], suggesting that the elevated proportion of essential genes in *S. pombe* may be the result of differential conservation of nonessential and essential genes in a reduced genome. However, *C. neoformans* also has a higher proportion of essential genes than *S. cerevisiae*, despite having more genes. This could be explained by the whole genome duplication in *S. cerevisiae*, which generated many redundancies in gene function [68,69]. Alternately, the TN-seq approach may be more liberal in assigning essential status than traditional genetic approaches. Indeed, a TN-seq based approach in *C. albicans* suggests that 27.5% (1,610 of 5,893 genes) of the genome is essential [32]. Because of the competitive growth in pooled settings, mutants that are sufficiently sick are difficult to sample, despite being viable, resulting in overestimation of the number of essential genes.

Practically, scoring essentiality as a binary phenotype, where mutants either grow or do not, may not be a totally mean-ingful distinction, particularly in the context of antifungal drug development. Severely impairing pathogen growth may not be dramatically different from completely impairing pathogen growth in terms of clinical outcome. Further, TN-seq may also score genes required for various growth stages as essential. Our mutant library preparation method required growth on synthetic defined media containing galactose and lacking uracil. Genes required for growth in these conditions may score as essential using TN-seq but not in traditional approaches. Indeed, we showed that some discrepancies between

PLOS Biology

the existing deletion collection and our TN-seq approach were explained by growth defects under these conditions. Essentiality is frequently context dependent, with a gene's essentiality being dependent on both the environment (e.g., plates versus mice versus human hosts) and on the genetic background of the pathogen. Future experiments exploring variation in essentiality in animal models of infection will likely be informative.

### Fluconazole resistance and tolerance are highly multigenic

Our TN-seq approach revealed numerous modifiers of fluconazole susceptibility and resistance. One of the strongest phenotypes we observed in our fluconazole TN-seq assay was for sensitivity to fluconazole conferred by loss of components of the *RIM101* pathway. This result was confirmed by analysis of individual mutants. The *RIM101* pathway is responsible for pH sensing and cell wall remodeling in response to elevated pH, as typically occurs upon entry into mammalian lungs. Here we find that the *RIM101* pathway function promotes resistance to fluconazole. This is not unprecedented as *RIM101* pathway components are also required for fluconazole tolerance in *Candida albicans* [70]. However, the Rim pathway in *C. neoformans* is substantially diverged from that found in the Ascomycete lineage, which includes *C. albicans*. In Ascomycetes, pH sensing is carried out by the Rim21 pH sensor acting in conjunction with the arrestin-like Rim8 and the Rim9 chaperone. None of these three proteins are present in Basidiomycetes and are instead replaced by a different upstream pH sensing module including Rra1 and Nap1 [59,71]. Given this dramatic upstream reprogramming of the Rim pathway in *C. neoformans* relative to the ascomycete *C. albicans*, it is interesting that a similar phenotype is present. In *S. cerevisiae*, the Rim21 sensor appears to respond to imbalances in plasma membrane lipids [72]. While Rra1 is not a homolog of Rim21, perhaps it is detecting and responding to the same type of plasma membrane stress in *C. neoformans*, including potentially that triggered by fluconazole, which impairs the production of the membrane sterol ergosterol. In fact, loss of the Cdc50 lipid flippase changes how Rim pathway activation occurs in *C. neoformans*, further supporting this model [73]. *cdc50Δ* mutants have increased fluconazole susceptibility [74] and our assay confirmed this, albeit with a weak *p*-value resulting from very few coding inserts in *CDC50* ($p = 0.035$, $n = 4$).

We also saw increased fluconazole resistance when transposon insertions were found upstream of genes involved in the electron transport chain and mitochondrial metabolism. This is consistent with past reports that rotenone, an inhibitor of the electron transport chain complex I, is very strongly antagonistic with fluconazole [61]. It is not immediately clear why downregulating electron transport chain function would make cells grow better in the presence of fluconazole. One possible explanation is that slowing growth may allow cells without complete inhibition of *ERG11* to produce enough ergosterol to prevent depletion from the membrane, thus allowing slower but more successful growth to continue [75]. Alternately, Acetyl-CoA is the primary input to both the TCA cycle and the first step of ergosterol synthesis. Slowing of mitochondrial metabolism could allow this metabolic precursor to shift from usage in the TCA cycle to usage for ergosterol synthesis, resulting in faster ergosterol synthesis even with Erg11 inhibition. Both models will be interesting to explore.

### High throughput genomics

TN-seq is one of several modern approaches that can be used to quickly assay gene function on a large scale. Broadly, these approaches fall into either insertional mutagenesis approaches (such as TN-seq) or CRISPR-based approaches. Many of the CRISPR approaches operate similarly to TN-seq except that they use a library of guide RNAs and a Cas9 to generate a single break and mutation per cell. The template sequence expressing the guide RNA can then be sequenced to determine which gene was disrupted. CRISPR approaches typically require synthesis of guide RNA libraries, which can be expensive, but has advantages in customization relative to TN-seq approaches. As well, in the short term, TN-seq is typically cheaper because of the lower startup costs, but in the long run CRISPR will typically become less expensive because of lower sequencing and library preparation costs for more targeted libraries with less complicated library preparation. However, CRISPR libraries gain this reduced cost by relying on existing annotation information. TN-seq

is annotation independent and thus allows assays of unannotated genes or regions that would likely be missed with a CRISPR library.

TN-seq also has significant advantages in portability. Our approach only requires the ability to introduce and mobilize a marked transposon into a haploid genome. In contrast with CRISPR-based approaches, transposon mutant libraries do not require the synthesis of expensive whole genome guide RNA libraries that may need to be modified for every additional species or even isolate in species with particularly high sequence divergence. Our TN-seq approach could be readily applied to additional isolates or species across the *Cryptococcus* pathogenic species complex. Variation in essential genes is common across various organisms, including fungi [27,76]. This variation also extends to drug resistance phenotypes as well [77]. Because this type of variation can be driven both by sequence polymorphism and by gene content polymorphism, future exploration of strain variation in *C. neoformans* and other fungi would likely benefit from a pangenome oriented approach, as is becoming common in bacteria [76]. Indeed, initial studies in fungi like *Aspergillus fumigatus* have revealed enormous variation in gene content between individual isolates that TN-seq approaches would be extremely powerful in exploring [78,79].

### Assaying essential gene function

Because deletion alleles of essential genes are inviable, screens using traditional deletion collections are unable to assay the contribution of essential genes to a given phenotype. This has long been a challenge for high throughput genetic assays and a number of solutions have been developed to allow assays of essential genes. These solutions generally rely on partial loss of function and include the construction of collections of repressible alleles [34,80], hypermorphic alleles including the DAmP (decreased abundance by mRNA perturbation) libraries [81], RNAi libraries [82,83], and CRISPRi libraries [84,85]. All of these approaches require either laborious one by one construction of targeted alleles (repressible promoters, DAmP alleles) or expensive synthesis of whole genome targeting libraries (RNAi, CRISPRi). In contrast, our TN-seq approach assayed 5′ modifiers across the entire genome in a single experiment that required no modifications to the typical TN-seq assay. We were able to identify modifiers of both the known essential gene and target of fluconazole *ERG11* as well as recapitulate previously described fluconazole interactions that were unveiled using chemical screens. TN-seq in combination with chemical inhibitors can help to reveal mechanism of action and predict potential chemical interactions to help guide therapeutic development. While our TN-seq assay was conducted in a pool, it can also provide guidance to construct individual mutants, as we did to generate the fluconazole sensitive *erg11* hypomorph (Fig 5D). Alternately, the pools can be cloned out and decoded using approaches like knockout sudoku to efficiently construct libraries of hypomorph alleles if desired [86]. As described above, this approach also benefits from portability, allowing TN-seq to potentially be applied to assay the function of essential genes across multiple isolates or species.

## Materials and methods

### Media

We routinely grew *Cryptococcus neoformans* strains on YPD media at 30°C. We made YPD using 20 g/L Bacto Agar, 20 g/L Bacto Peptone, 10 g/L Bacto Yeast Extract and 2% glucose. We made YPD liquid the same way but omitting agar. We supplemented media with G418 (NEO) at 200 µg/mL and Nourseothricin (NAT) at 100 µg/mL. We made SC-URA media using 6.7 g/L of yeast nitrogen base without amino acids, 20 g/L Bacto Agar, 2% glucose and 1.92 g/L SC-URA dropout powder mix. For raffinose or galactose media, we substituted the 2% glucose with 2% raffinose or 2% galactose, respectively. Strains were cryopreserved in 20% glycerol.

We routinely grew bacterial stocks at 37°C. We made LB plates with 5 g/L Bacto Yeast Extract, 10 g/L Bacto tryptone, 10 g/L sodium chloride, and 15 g/L bacto agar. We supplemented media with 100 mg/L ampicillin or 25 mg/L kanamycin. Strains were cryopreserved in 20% glycerol.

## Strain construction

We generated strains using an electroporation protocol modified from [87]. We grew strains to saturation overnight at 30°C in YPD and diluted back to an $OD_{600}$ of 0.2 in 100 mL of fresh YPD liquid. We then grew with shaking at 30°C until cultures reached an $OD_{600}$ between 0.6 and 1.0. We then centrifuged cells to pellet them and washed them twice with ice-cold water. We resuspended them in 10 mL of electroporation buffer (10 mL Tris-HCl (pH 7.5), 1 mM $MgCl_2$, 270 mM sucrose, 1 mM DTT). We incubated cells on ice for one hour. We then pelleted cells again and resuspended them in 250 μL of EB. We mixed 45 μL of cell suspension with 5 μL of transformation DNA in a pre-cooled 2 mm gap electroporation cuvette. We then transformed using a BioRad Gene Pulse with settings of 0.45 kV and 125 μF. Immediately after electroporation, we added 2 mL of liquid YPD and transferred cells to a round bottom culture tube. We let cells recover overnight at room temperature and then pelleted cells to plate on selective media. After colonies were visible, we streak purified colonies on selective media before preparing DNA to PCR validate transformants.

We generated our TN-seq library strain (yBB119) in two steps. We first commercially synthesized (IDT) a Ds mini-transposon [88] containing a NEO resistance marker [41] (pBB51). We then used overlap PCR to add homology to the up and downstream regions surrounding an intron of *URA5.* We performed a first round PCR to build upstream and downstream regions with homology to a product containing the transposon (oBB229 and oBB283, 5′; oBB285 and oBB286, transposon; oBB284 and oBB235, 3′). We then used a second round PCR to assemble the pieces (oBB229 and oBB235). We gel extracted the amplified band and topo blunt cloned to make plasmid pBB65. We then digested plasmid pBB65 with KpnI and NotI to release the transposon with targeting homology and used that as template for a transformation into H99 *C. neoformans* to produce strain yBB115. This strain was selected on G418 to select for transposon integration. G418 resistant colonies were checked for growth on SC-URA to detect integrations in the correct location.

We also commercially synthesized (IDT) a codon optimized version of the hyperactive AcTPase4x (Ac) transposase (pBB52) [89]. We codon optimized the Ac transposase using the Optimizer webtool and the "random" method which assigns codon usage probabilistically across the gene sequence to roughly match the codon usage of the genome [40]. We added a GAL7 promoter using overlap PCR. In our first round PCR we amplified the GAL7 promoter (oBB183 and oBB290) and the Ac transposase (oBB271 and oBB272). We then assembled these pieces in a second round PCR (oBB183 and oBB272). We gel extracted the amplified band and topo blunt cloned this product to make plasmid pBB63. We then subcloned the gal driven Ac into plasmid pSDMA25 [41] by digesting both vectors with XhoI. We additionally digested pBB63 with XmaI to digest the vector backbone to produce bands of a different size from the released insert. We called the resulting plasmid pBB68. We digested this plasmid with PacI to linearize it and transformed it into yBB115. We selected transformants on YPD + NAT plates. We verified transposase activity by checking transformants for the ability to produce colonies on SC-Ura plates after overnight growth in YPGalactose liquid media and conducted a stability assay to check for stable integration. A successful strain was frozen down as yBB119.

To generate specific transposon inserts at the 5′ end of *ERG11*, we used overlap PCR to add homology arms to the Ds transposon. To generate 5′ inserts that disrupted *ERG11* function (*erg11–77*), we amplified the upstream region from H99 genomic DNA using primers oBB583 and oBB595, the downstream from H99 genomic DNA using primers oBB598 and oBB588, and the transposon from pBB51 using oBB596 and oBB597. We then stitched together the pieces in a second PCR using primers oBB589 and oBB590. We excised the amplified band from a gel, blunt cloned it with a topo blunt kit and sequence verified as pBB102. To generate inserts that did not disrupt *ERG11* function (*erg11–296*) we amplified the upstream region from H99 genomic DNA using primers oBB583 and oBB591, the downstream from H99 genomic DNA using primers oBB594 and oBB588, and the transposon from pBB51 using primers oBB592 and oBB593. We then stitched the pieces together in a second PCR using primers oBB589 and oBB590. We excised the amplified band from a gel, blunt cloned it with a topo blunt kit and sequence verified it as pBB100. We digested each plasmid with XhoI to release the insert and transformed into H99. We verified transformants using stability tests and PCR validation for junction PCRs

(oBB212 and oBB583, 5′ junction; oBB171 and oBB588, 3′ junction). We called the resulting strains yBB163 and yBB164 (pBB100, *erg11–296*) and yBB165 and yBB281 (pBB102, *erg11–77*).

To generate yBB407 and yBB408, we used overlap PCR to add homology arms to the Ds transposon. We amplified the upstream region from H99 genomics DNA using primers oBB1173 and oBB1176 and the downstream region from H99 genomic DNA with primers oBB1175 and oBB1177. We then PCR amplified the transposon from pBB51 using the upstream and downstream homology regions as primers and with the addition of nested primers oBB1172 and oBB1174. We also amplified guide RNAs in a stepwise process, using primers oBB1171 and oBB961 to amplify the U6 promoter from JEC21 DNA and primers oBB964 and oBB1170 to amplify the gRNA scaffold from plasmid pDD162 [90]. We then used overlap PCR to combine these two pieces using primers oBB962 and oBB963. We then amplified Cas9 from plasmid pBHM2403 [91] with primers M13F and M13R. We transformed all three pieces of DNA into H99 using electroporation. We validated transformants using PCRs spanning the insert with primers oBB1173 to oBB1175. Strains yBB407 and yBB408 resulted from independent transformations started from different cultures. Strain, plasmid, and primer lists are available as S5–S7 Tables.

## Mutant library construction

We constructed our TN-seq mutant library from strain yBB119, which was built as described above. We began library construction by subculturing yBB119 from the freezer onto YPD + 5-FOA plates. We then started 5 × 10 mL cultures of yBB119 in YPD + 5-FOA liquid media from independent colonies. We shook these cultures overnight at 30°C. We then centrifuged the cultures, decanted the supernatant, and resuspended the cell pellet in 20 mL of YP + raffinose. We then shook them overnight again at 30°C. The next day we spun down, decanted, resuspended in water, spun down a second time, decanted, and finally resuspended in 100 mL of YPGal. We then plated 500 μL of culture to 160 SC-URA + GAL plates. With 5 replicate cultures this totaled 800 plates. We incubated the plates for 4 (3 sets of plates) or 5 days (2 sets of plates) before scraping the colonies off of the plates using water and a cell spreader. We then spun them down, washed once with water and then resuspended in 500 mL of water. We then diluted that culture 1:10 into SC-URA media, which we shook overnight at 30°C. We froze 1.5 mL aliquots from these cultures in 20% glycerol.

We then revived all five cultures from 1.5 mL aliquots in 50 mL of YPD + G418. We allowed them to grow to saturation with shaking at 30°C and then mixed all five cultures to form a pool. We froze 1.5 mL aliquots of this library in 20% glycerol in cryovials for future experiments.

## TN-seq sequencing library preparation

To prepare DNA for sequencing, we used a modified version of the Qiagen Genomic-Tip Yeast DNA protocol. We followed all standard steps except that we extended both the lyticase and proteinase K treatments to 24 h on a shaking incubator. Prior to applying DNA to the genomic-tip columns, we transferred the digested cells to a 50 mL tube containing lysing matrix Y (MP Biomedicals). We then freeze cracked twice in liquid nitrogen (alternating between liquid nitrogen and a 42°C water bath). We then bead beat the cells using a Fastprep big prep adapter with 40 s of shaking at 4.0 m/sec with 5 min pause in between. We then pelleted the cells and applied the supernatant to the Qiagen genomic tip column.

We prepared TN-seq libraries as previously described in *S. pombe*, with some modifications [25] (S8 Fig). We digested purified DNA using either MseI (10,000 U/mL) or CviAII (10,000 U/mL). Our digests used 10 μg of DNA in 500 μL of volume overnight at 37°C in CutSmart buffer, including 15 μL of each enzyme. We then used SPRIselect beads to clean and size select the digested DNA. We washed with 0.5 volumes of beads, pelleted on a magnet, removed the supernatant and added 0.2 volumes of beads before precipitating again. We then washed and eluted the DNA with 500 μL of water. We then added linkers with unique random barcodes via end ligation (88 μL of 10× ligation buffer, 143.5 μL of water, 153 μL of annealed barcoded linker, 5 μL of T4 DNA ligase, and 490 μL of the cleaned and size-selected DNA). We made the linkers from primers oBB912 (MseI) or oBB913 (CviAII) combined with primer oBB914. We mixed the primers at a concentration

of 10 µM each in 1× HF PCR buffer and denatured at 95°C for 1 min, followed by cycles of 10°C decreases in temperature for 7 min until a final cycle at 20°C.

After end ligation, we split the DNA from each digest into two pools. We used PCR to amplify these pools separately with 4 different Ds-specific oligos (oBB915, oBB916, oBB917, oBB918), intended to create diversity in starting location. We thus performed 12 PCR reactions per digest per pool, except for oBB918, for which we performed only 9 PCR reactions per digest. In addition, we ran 3 linker-only controls per pool. In sum, each digest thus produced two pools of 45 reactions, or 90 reactions per digest. Each PCR reaction contained 24.5 µL water, 10 µL 5× HF buffer, 1 µL dNTPs (10 mM), 8 µL linker ligated DNA, 1 µL linker oligo oBB919 (10 µM), 5 µL Ds-specific oligo (2 µM) or water, and 0.5 µL Phusion polymerase (2,000 U/mL). We ran this PCR using 94°C for 1 min, 6 cycles of (94°C for 15 s, 65°C for 30 s, 72°C for 30 s), 24 cycles of (94°C for 15 s, 60°C for 30 s, 72°C for 30 s), and finally 72°C for 10 min.

We then combined the reactions with shared primers from each pool (12 or 9 reactions) to form a subpool. We ran a subset of these subpools on a gel for validation. Successful amplification of a TN-seq library produces a smear of DNA rather than discrete bands. Defined bands suggest low complexity libraries, potentially resulting from jackpot events.

We then added the remaining Illumina adapters and barcodes with a second round of PCR on the subpools. This second reaction contained 28.5 µL water, 10 µL 5× HF buffer, 1 µL dNTPs (10 mM), 2.5 µL oBB920 (10 µM), 2.5 µL barcode oligo (oBB921-oBB932; 10 µM), 5 µL of insert PCR pool, 0.5 µL Phusion polymerase (2,000 U/mL) and was incubated at 94°C for 2 min, 5× (94°C for 30 s, 54°C for 30 s, 72°C for 40 s). We gave each individual pool a unique barcode (1 barcode per enzyme digest mix). For each subpool, we ran 4 PCRs or 3 for those derived from oBB918. We then pooled reactions these reactions and cleaned the DNA using SPRIselect beads. We used 600 µL of pooled PCR product from the reactions above and added 0.75 volumes of SPRI beads. We precipitated the beads on a magnet, removed the supernatant and washed the pellet with 85% ethanol. We then eluted with 200 µL of TE. We repeated this wash a second time.

Finally, we confirmed this library by using a PCR with oligos that matched the end of the Illumina adapters (oBB933 and oBB934). This reaction was 15.75 µL water, 5 µL 5× HF buffer, 0.5 µL dNTPs (10 mM), 1.25 µL oBB933 (10 µM), 1.25 µL oBB934 (10 µM), 1 µL library DNA, 0.25 µL Phusion polymerase (2,000 U/mL). We ran this PCR at 98°C for 30 s, 35× (98°C for 10 s, 62°C for 30 s, 72°C for 40 s), 72°C for 10 min. We then quantified the library using a Qubit and determined fragment sizes using a Bioanalyzer. Sequencing was performed on a NextSeq instrument at the Stowers Molecular Biology Core. Raw sequencing reads are available at PRJNA1134100 on the Sequence Read Archive.

**TN-seq fluconazole assay**

To perform our fluconazole assay, we revived a 1.5 mL aliquot of our TN-seq library in 99 mL of YPD. After it grew to saturation, we inoculated two 100 mL YPD cultures at $OD_{600}$ = 0.1. We then saved the remaining culture to prepare DNA for a $T_0$ time point. We added 0.1 mL of DMSO to one culture and 0.1 mL of DMSO containing 13.5 mg/mL of fluconazole to the other. We grew these cultures with shaking at 30°C overnight. The next day we diluted back to $OD_{600}$ = 0.1 in 100 mLs of fresh YPD with 0.1% DMSO and with or without fluconazole. We grew to saturation overnight with shaking at 30°C again. The next day we collected cell pellets via centrifugation to prepare DNA.

**Data analysis**

We processed data by first identifying reads with exact matches to the Ds transposon sequence with a custom perl script (available at https://github.com/bbillmyre/Crypto_TNseq and 10.5281/zenodo.15297569). We trimmed Ds and linker sequences, saving the barcode sequences trimmed from the linker. We then mapped the reads to the H99 genome (v44 from FungiDB [49,50]) using bwa mem [92]. This approach allows multiple mapping of reads so that reads mapping to repetitive regions are allowed to map, albeit with very low mapping scores. We used samtools [93], picard, and bedtools [94] for further processing before identifying the start of each transposon derived read and outputting a .bed file using a custom perl script (available at https://github.com/bbillmyre/Crypto_TNseq and 10.5281/zenodo.15297569). We reduced

inserts that shared a common linker barcode to a single representative. We then used $R$ (v.4.3.1) [95] to combine bed files by site from the various runs ($T_0$, DMSO, and $IC_{50}$ fluconazole). We converted insert counts to insertion frequencies and set 0 insert sites (where there were inserts at $T_0$) from the DMSO and $IC_{50}$ fluconazole libraries to the minimum frequency >0 otherwise detected in that library. Resetting 0 values to minimum frequency artificially reduces the size of the phenotypes detected but allows log adjustment of our data. We also removed any sites with less than 6 inserts in the original $T_0$ pool to reduce potential sampling error in the fluconazole experiments. We then labeled each insert site either with the gene body it was part of or as intergenic if not between an ATG and STOP codon of a gene.

For our machine learning model, we then used custom $R$ scripts to calculate parameters describing the landscape of transposon insertions across each gene (available at https://github.com/bbillmyre/Crypto_TNseq). For each gene we determined the following: chromosome, gene length, total number of inserts, total number of inserts divided by gene length, sum of insert frequency, sum of insert frequency divided by gene length, largest insert free gap in bases, largest insert free gap as percentage of gene length, sum of insert frequency in the middle 80% of the gene body, sum of insert frequency in the middle 80% of the gene body divided by length, and sum of insert frequency over the surrounding 100 kb sequence surrounding the gene. We obtained a set of gene essentiality and orthology predictions for *C. albicans*, *S. cerevisiae*, and *S. pombe* from a previous study [32]. We used FungiDB to find orthologs in *C. neoformans* of the *C. albicans* genes in this dataset. We then selected a set of genes with orthologs in all four species that shared essentiality status in *C. albicans*, *S. cerevisiae*, and *S. pombe*. We used this set as a training set for a machine learning model. Using $R$, we randomly split it into 80% training data and 20% validation data (random seed = 100). We then trained using the "randomForest" package with parameters (mtry = 5, ntree = 10,000, importance = TRUE)[96]. We repeated this 100 times, saving importance parameters and precision/recall estimates from the validation set for each run. We also used each model to predict the genome each time.

We then used the variation across these models to test whether the mean essentiality score (ES) was statistically different from the 0.5 cutoff (Student $t$ test). As a result, we define three classes of genes: essential genes (ES > 0.5 and $p < 0.05$ after Bonferroni correction), nonessential genes (ES < 0.5 and $p < 0.05$ after Bonferroni correction) and unclassified genes ($p > 0.05$). Using this approach, we predicted 1,412 essential genes, 5,382 nonessential genes, and 181 unclassified genes. We then revisited our training data using these classifications. Of 1,653 conserved genes in our original training set, 89 were unclassified (5.4%), compared with just 92 in the remaining 5,322 genes (1.7%) ($p < 0.0001$, Fisher's exact test). Given that the original assignment of essentiality was based on conservation, we were concerned that our model may have been overfit and was struggling to accurately determine essentiality for genes in the training set whose actual essentiality did not match the original assumption. Indeed, we identified no genes in our training set that were statistically assigned opposite essentiality from the initial assumption we provided from homology.

We manually reviewed transposon insertions in many of these genes, which suggested that the model was struggling to predict essentiality for these genes because the TN-seq data disagreed with the conservation-based assumptions that were part of the training set. We thus retrained our model using a training set without all 89 genes with unclassified essentiality in our first model. We dropped all of the genes with $p > 0.05$ that were part of the original training set and repeated our model construction as above to produce our final model. As before, we predicted essentiality from this model 100 times. Our revised model had just 31 unclassified genes in the training set (2.0%) compared with 60 in the remaining 5,411 (1.1%). While this difference was still statistically significant ($p = 0.011$, Fisher's exact test), the magnitude of the discrepancy was substantially reduced from the first model.

For the fluconazole experiments we used $R$ to perform statistical tests. We calculated a $log_{10}$ fold change for $IC_{50}$ to DMSO growth (mathematically the same as the ratio of $IC_{50}/T_0$ to DMSO/$T_0$). We used a Mann–Whitney $U$ test to compare the distribution of fold-changes in a gene region to that of the intergenic inserts. We also performed multiple test correction using a Bonferroni correction. For 5′ and 3′ modifiers we performed the statistical tests in exactly the same manner except that we compared the distributions of fold changes of inserts within the 300 bp immediately upstream of the start codon (3′) or downstream of the stop codon (5′) to the intergenic inserts. We again performed a Bonferroni correction.

## Competition assay

To conduct competition assays, we grew cultures overnight in YPD at 30°C. We then added enough KN99 and enough competitor strain to reach an $OD_{600}$ of 0.05 of each strain to 5 mL of SC-URA + GAL broth. We plated a concentration series of these initial mixed cultures to YPD plates. We then grew the cultures overnight with shaking at 30°C. The next day we plated a concentration series to YPD plates. After colonies grew from each concentration series, we replica plated to YPD + NEO and to new YPD plates. We counted colonies the next day and recorded the number of colonies that grew on YPD + NEO (just marked strain) versus YPD (both strains).

## Spot dilution assays

To perform spot dilution assays, we grew cultures overnight in YPD at 30°C until they reached saturation. We then measured the $OD_{600}$ using a spectrophotometer. We diluted cells to an $OD_{600}$ of 20. We then performed 10-fold serial dilutions and plated 5 μL spots of each dilution to the appropriate media. Plates were incubated at 30°C until we took pictures as detailed in each figure.

## Growth curves

To measure growth curves, we grew cultures overnight in YPD at 30°C until they reached saturation. We diluted cells to an $OD_{600}$ of 0.00166 in fresh YPD and added 150 μL to each well of a 96 well plate. We then made a drug stock containing (for each sample) 50 μL of YPD with 0.2 μL of DMSO containing 1000X fluconazole. This stock was thus 0.4% DMSO and 4X fluconazole for each condition. We then added 50 μL of drug stock YPD mix to each well to achieve a final mix in each well of cells at an $OD_{600}$ of 0.00125, DMSO at 0.1%, and fluconazole (if present) at 1X concentration. We grew the plates on a TECAN Infinite M200 Pro at 30°C with shaking. We ran 8 technical replicates for each sample and report the $OD_{600}$ at 24 h normalized to the no drug control at 24 h.

## Supporting information

**S1 Fig. Transposon insertions are biased regionally and by selection. (A)** Transposon insertion count is plotted across chromosomes. There are clear peaks at the rDNA array on Chromosome 2 and the *URA5* locus on Chromosome 8, as well as the surrounding chromosome. **(B)** Transposon insertion count is plotted across the 2000 bases immediately upstream and downstream of the start codon. Inserts within the gene bodies are distributed among equally sized windows for each gene. Rows are split based on predicted gene essentiality. **(C)** Insertions per base are plotted for genes not present versus present in the deletion collection. Boxplots show first quartile, median, third quartile. The whiskers show the range to a maximum of 1.5 times the interquartile range above and below the first and third quartile, respectively. Outliers are displayed as individual datapoints. Data underlying **A** and **B** can be found in S1 Data at 10.5281/zenodo.15264486. Data underlying **C** can be found in S1 Data at 10.5281/zenodo.15264486 and in S1 Table.
(TIF)

**S2 Fig. PCR validation of predicted essential genes with strains in deletion collection. (A)** Schematic of PCR validation strategy for deletions. WT alleles should be detectable with an in-gene PCR that amplifies wildtype DNA sequence (left). Mutant alleles should have the wildtype allele replaced with a drug resistance marker. The junctions between the genomic sequence and the resistance marker should be detectable via PCR on both the 5′ and 3′ ends (right). A successful mutant should produce bands from the 5′ junction and 3′ junction but not from the in-gene PCR. A wildtype strain should produce a band only from the in-gene PCR. **(B)** PCR validations for 5 independent colonies from the deletion collection strain for CNAG_05292. All five produce wildtype in-gene bands and no junction bands. **(C)** PCR validations for 5 independent colonies from the deletion collection strain for CNAG_06887. All five produce wildtype in-gene bands and no junction bands. **(D)** PCR validations for deletion collection strains for CNAG_04763, CNAG_00996, CNAG_02190. All

three strains produce negative in-gene PCRs and successful junction PCRs for both 5′ and 3′ ends. Original gel images can be found at 10.5281/zenodo.15264486 as S1 Raw images.
(TIF)

**S3 Fig. Some incorrectly predicted essential genes have growth defects in transposase inducing conditions. (A)** Spot dilution assays with 5 μL spots plated. The initial leftmost spot is of $OD_{600}$ = 20 culture and each successive spot is a 10-fold dilution, so that the final spot should be $10^5$ less concentrated than the first. All four plates were spotted on the same day with the same dilution series. **(B)** Competition assay with percentage of mutant plotted on the *y*-axis. Each mutant was competed against the same unmarked wildtype KN99 parental strain. Strains were competed in the SC-URA + Gal media used in the original assay. Strains were originally mixed at a 50:50 ratio based on $OD_{600}$. Inconsistency with mixing of CNAG_00996 suggests an altered $OD_{600}$ to CFU ratio. **(C)** Picture of colonies on YPD plates from the competition assay for the *cnag_00996Δ* mutant. Colonies were distinctly different in appearance and replica plating to YPD + NAT media confirmed that rough colonies were the mutant strain. Panel on right is zoomed in from inset box on left and is adjusted to help visualize difference between colonies more clearly. Original images underlying **A** and **C** can be found in the Stowers Original Data Repository at http://www.stowers.org/research/publications/libpb-2480. Data underlying **B** can be found at 10.5281/zenodo.15264486 as S3B Fig.
(TIF)

**S4 Fig. Concentration dependent inhibition of growth by fluconazole in YPD liquid.** Growth at 24 h plotted relative to a no drug control. *X* axis displays concentration of fluconazole added in DMSO. *Y* axis shows $OD_{600}$ normalized to $OD_{600}$ of no drug control. The underlying data can be found at 10.5281/zenodo.15264486 as S4 Fig.
(TIF)

**S5 Fig. Insert orientation plays little role in downstream effects.** Boxplot displaying distribution of $log_{10}$-adjusted fold changes in insert density (i.e., frequency in fluconazole/frequency in DMSO). Each column shows inserts only within a 300 base pair region either immediately upstream of the start codon or downstream of the stop codon. Boxplots show first quartile, median, third quartile. The whiskers show the range to a maximum of 1.5 times the interquartile range above and below the first and third quartile, respectively. Outliers are displayed as individual datapoints. Inserts are split based on whether they are oriented in the same direction as *ERG11* (plus) or the opposite orientation (minus). Underlying data can be found in S1 Data at 10.5281/zenodo.15264486.
(TIF)

**S6 Fig. *cnag_01137* −155 allele confers resistance to elevated doses of fluconazole.** Spot dilution assays with 5 μL spots plated. The initial leftmost spot is of $OD_{600}$ = 20 culture and each successive spot is a 10-fold dilution, so that the final spot should be $10^6$ less concentrated than the first. All plates were spotted on the same day with the same dilution series. YPD plates were imaged after 48 h at 30°C and fluconazole plates were imaged after 72 h at 30°C. Original images can be found in the Stowers Original Data Repository at http://www.stowers.org/research/publications/libpb-2480.
(TIF)

**S7 Fig. Shiny app allows visualization of data on publicly available website.** Screenshot of a publicly available interactive Shiny app (https://bbillmyre.shinyapps.io/Crypto_TN_seq_viewer/) that visualizes data from the *C. neoformans* TN-seq assay. There are four plots, displaying the distribution of fold changes within a gene compared with intergenic inserts (as in Fig 4C,) the distribution of insert frequencies across a gene at three different experimental stages, transposon insertion frequencies across a gene with an additional 300 bases before the ATG and after the stop codon, and finally a volcano plot (as in Fig 4B) with the current gene highlighted in black. The app only accepts *C.*

*neoformans* systematic names (i.e., CNAG_0####). Underlying data can be found in S1 Table and in S1 Data at 10.5281/zenodo.15264486.
(TIF)

**S8 Fig. Diagram of library preparation approach. (A)** Diagram showing sequential digests and adapter ligation/PCR to amplify the boundary between the known Ds sequence and the unknown genomic sequence. Notably, primers cannot amplify between the two ligated adapters because the ligated adapters are modified so that they can only serve as the template in a second-round reaction. **(B)** Zoom in of primers amplifying the transposon insertion. The four Ds specific primers anneal to the same location but have varying lengths so that the products produced will have a diversity of starting nucleotides when sequenced.
(TIF)

**S1 Table. Table containing essentiality predictions and fluconazole response data.**
(XLSX)

**S2 Table. Table containing GO predictions for predicted essential genes lacking predicted human orthologs.**
(XLSX)

**S3 Table. Table containing genes predicted essential in all fungi but lacking human orthologs.**
(XLSX)

**S4 Table. Table containing essentiality predictions for genes with orthologs in other fungi with TN-seq datasets.**
(XLSX)

**S5 Table. Strain table.**
(XLSX)

**S6 Table. Plasmid table.**
(XLSX)

**S7 Table. Primer table.**
(XLSX)

**S1 Image. Raw images** .
(PDF)

## Acknowledgments

We thank members of the Billmyre and Zanders labs for helpful comments on the manuscript. We thank the Heitman and Alspaugh labs for supplying strains. We thank the Madhani Laboratory for prepublication access to the *C. neoformans* knockout collection and acknowledge the NIH funding that has helped fund its construction (R01AI100272).

## Author contributions

**Conceptualization:** Sarah E. Zanders.

**Data curation:** R. Blake Billmyre, Amy M Kuhn.

**Formal analysis:** R. Blake Billmyre, Caroline J Craig, Joshua W Lyon, Claire Reichardt, Amy M Kuhn, Michael T Eickbush.

**Funding acquisition:** R. Blake Billmyre, Sarah E. Zanders.

**Investigation:** R. Blake Billmyre, Caroline J Craig, Joshua W Lyon, Claire Reichardt, Amy M Kuhn, Michael T Eickbush.

**Methodology:** R. Blake Billmyre, Caroline J Craig, Joshua W Lyon, Claire Reichardt, Michael T Eickbush.

**Project administration:** R. Blake Billmyre, Sarah E. Zanders.

**Resources:** R. Blake Billmyre, Sarah E. Zanders.

**Software:** R. Blake Billmyre, Claire Reichardt.

**Supervision:** R. Blake Billmyre, Sarah E. Zanders.

**Validation:** R. Blake Billmyre, Caroline J Craig, Joshua W Lyon, Claire Reichardt.

**Visualization:** R. Blake Billmyre, Joshua W Lyon.

**Writing – original draft:** R. Blake Billmyre.

**Writing – review & editing:** R. Blake Billmyre, Caroline J Craig, Joshua W Lyon, Claire Reichardt, Amy M Kuhn, Michael T Eickbush, Sarah E. Zanders.

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
