## [Editor Report · Decision Letter 0]

16 Aug 2024

Dear Dr Billmyre, 

Thank you for submitting your manuscript entitled "Saturation transposon mutagenesis enables genome-wide identification of genes required for growth and fluconazole resistance in the human fungal pathogen Cryptococcus neoformans" for consideration as a Research Article by PLOS Biology.

Your manuscript has now been evaluated by the PLOS Biology editorial staff and I am writing to let you know that we would like to send your submission out for external peer review.

Once your full submission is complete, your paper will undergo a series of checks in preparation for peer review. After your manuscript has passed the checks it will be sent out for review. To provide the metadata for your submission, please Login to Editorial Manager (https://www.editorialmanager.com/pbiology) within two working days, i.e. by Aug 18 2024 11:59PM.

Kind regards,

Melissa

Melissa Vazquez Hernandez, Ph.D.

Associate Editor

PLOS Biology

---

## [Decision Letter · Decision Letter 1]

4 Oct 2024

Dear Dr Billmyre,

Thank you for your patience while your manuscript " Saturation transposon mutagenesis enables genome-wide identification of genes required for growth and fluconazole resistance in the human fungal pathogen Cryptococcus neoformans " was peer-reviewed at PLOS Biology. It has now been evaluated by the PLOS Biology editors, an Academic Editor with relevant expertise, and by two independent reviewers, being reviewer #2 Henry Levin. 

In light of the reviewer reports included below, we would like to invite you to revise your manuscript to address the feedback provided. Reviewer #2, in particular, raised concerns about the study’s broad interest. Reviewer #1 requests testing cps25 on fluconazole plates and examining whether mitochondrial genes affect other antifungals. Additionally, Reviewer #2 suggests deleting specific mitochondrial genes to further support their role in fluconazole resistance, and requests a metagene plot to visualize the density of insertions in intergenic regions. We agree with these suggestions, which will require some additional experimental work, as addressing these concerns will significantly strengthen the manuscript. We also want to emphasize the importance of testing additional antifungals to broaden the study’s relevance.

Given the extent of revision needed, we cannot make a decision about publication until we have seen the revised manuscript and your response to the reviewers' comments. Your revised manuscript is likely to be sent for further evaluation by all or a subset of the reviewers.

**IMPORTANT - SUBMITTING YOUR REVISION**

*Re-submission Checklist*

*Published Peer Review*

*PLOS Data Policy*

*Blot and Gel Data Policy*

Sincerely,

Melissa

Melissa Vazquez Hernandez, Ph.D.

Associate Editor

PLOS Biology

REVIEWERS' COMMENTS

Reviewer #1: 

Cryptococcus is a critical priority human fungal pathogen (as designated by the WHO) and is very much understudied. It is rapidly becoming the model basidiomycete. This study provides a very important dataset, which will be extremely useful to the Cryptococcus community (of the essential genes including those genes that can only tolerate transposon insertion in their 5' UTR). The shiny app makes this dataset very accessible: I have tried this out for my 'favourite genes' and am very impressed with the data described. 

The study also provides an excellent example of the power and the many uses that can be made of the TN-seq technique. Overall the findings are novel and of significance to the Cryptococcus community and well beyond - for those interested in drug resistance, and those keen to try out this powerful technique in their own system. I am not an expert on machine learning and big data analysis, but the statistical analysis seems to me to be appropriate and I believe that all data needed to replicate the study are being made available here. The supplementary information will be very valuable to the Cryptococcus community.

The manuscript is well written and the data presented very clearly in the figures. As such, I am very supportive of the publication of this manuscript. Below, I suggest a few changes to be made before publication.

Figure 4: on page 28 it is pointed out that several components of the Rim1 pathway were identified amongst the hits for fluconazole sensitivity. The authors should briefly explain here what is known about the functions of this pathway (rather than having the reader wait for this information, which is provided in the discussion). Why was vps25 not tested on the fluconazole plate? This should be included for completion.

Figure 5: this is an important figure demonstrating a very useful facet of TN-seq mutagenesis. Many mitochondrial genes were identified here, suggesting that perturbed 'general' mitochondrial functions can lead to fluconazole resistance. However, I doubt this is specific to fluconazole resistance. The authors should test whether these strains are 'multi-drug resistant'. I think it likely that they have several stress response pathways activated and are possibly overexpressing several efflux pumps etc. The authors can refer to reviews and specific papers on this. eg. https://doi.org/10.3390/microorganisms8101574;
www.pnas.org/cgi/doi/10.1073/pnas.1911560116; doi:10.1128/EC.05184-11 (review).

Reviewer #2: 

TN-seq in Cryptococcus neoformans

Billmyre et al

The authors describe a comprehensive application of TN-Seq with Cryptococcus using machine learning to identify essential genes. They apply this method to identify genes involved in fluconazole resistance. In characterizing intergenic insertions, they find that the density of insertions upstream of genes can be used to characterize the contributions of essential genes that otherwise cannot tolerate insertions in their ORFs. 

This is a carefully conducted set of experiments combined with a scholarly description of the method. The quality of writing is high and the results well documented. And for the most part the interpretations are solidly supported by the data. The result is a valuable list of essential genes in Cryptococcus and a list of genes involved in fluconazole resistance. This data provides an important resource for future work on drug resistance. Although I find nothing of significance is wrong with this work, I must say that TN-Seq has been used on fungi for over 10 years including experiments to determine genes involved in fluconazole resistance. My view is this is a quality study but that it may not be of wide interest to the readers of PLOS Biology. As this is a study of quality genetics, I think it will be of high interest to readers of important genetic journals. 

Specific comments

* Please use page numbers and or line numbers. Its hard to refer to specific text without them. I will use the page numbers of the PDF as listed by Adobe Acrobat. 

* Abstract, line 11, We show that 5' insertions… I would specify insertions 5' of genes. 

* Page 13, 7 lines from bottom. Please provide reference support for statement about transmission. 

* Page 14, line 8. Please provide the total number of genes in Cryptococcus. 

* Page 14, 10 lines from bottom. There are at least two papers demonstrating deletion sets have a high frequence of calling essential genes nonessential. Deletion strains carry aneuploids and rearrangements. Guo et al 2013, and PMID: 10888885.

* Page 14, 6 lines from bottom. There are other references of TN-seq to include. Cerevisiae study of prion pathology PMID: 29769283, Galbrata PMID: 32819971, Yarrowia PMID: 29792930, Albicans study with Azole resistance PMID: 30374049.

* Page 14, second line from bottom. Should comment that TN-seq requires haploid genomes. 

* Page 15, line 2. there are a variety of computational approaches. You may want to compare the advantages of machine learning. A list of computational approaches is provided in PMID: 32886545.

* Page 15, line 8. The limitations of CRISPR include lack of information about essential genes. And regions/genes not annotated correctly cannot be tested. TN-Seq can identify intergenic and noncoding sequences that have function.

* Page 15, bottom. 3rd line from bottom. Not the first use of TN-seq to query essential gene function. The structure of this sentence confuses how all TN-seq studies identify essential genes and how some can query specific function of essential genes. Lee et al measured contributions of essential genes to heterochromatin function. And Mitchel et al studied individual domains in essential proteins by TN-seq. Best to modify your claim to focus on upstream insertions. 

* Page 16, line 8. Typo, unlikely capable of splicing…

* Page 17, line 6. This statement describes selection for excision not transposition. G418 is used to select for insertions. I would mention that here. Also, I would like to see more detail in your methods regarding the size of the random barcode and the the PCR primers to know the distance between the PCR primer sequence and the random barcodes. perhaps a drawing in the supplement would help.

* Page 17, 4 lines from bottom. Local hoping is a very interesting behavior that has been observed for many TEs including Hermes, Ds, sleeping beauty, piggybac. P etc.

* Page 17, Suppl. Fig 1B. This is a very accurate description of insertions across the start codon. But it would also be very interesting to see a metagene plot across genes. Hermes in pombe and cerevisiae showed higher level of integration on either side of the ORF due to nucleosome distribution.

* Page 18, line 5. is this kb of the entire genome or of the mappable genome minus repeat sequences, rDNA. 

* Machine learning section. I would like to commend the authors for their scholarly description of the machine learning methods. This builds confidence in the predictions. I would suggest that their first failed approach that trained with the 89 unknown genes be moved to the supplement. 

* Page 18, 9 lines from bottom. In addition to ref 36, PMID: 32886545 contains a (at the time) comprehensive list of TN-seq experiments including in eukaryotes. 

* Page 19, 3rd line from bottom. Remind the reader the 1,653 genes are the conserved set. 

* Page 20, 8 lines from bottom. I would find better language than unknown genes. Its their essentiality that is unknown. 

* All Supplemental Tables. They really need legends that describe their content. Can be in a separate page of the excel files. 

* Page 22, last line. including the absence of an in-gene PCR product.

* Page 23, 10 lines from bottom. Did you find genes required to induce the Gal promoter. Wouldn't these be misclassified as essential? Discuss here?

* Page 24, 11 lines from bottom. This is a valuable list for future study. Will the authors discuss a subset of best to target. 

* Page 25, 1st line. These 34 genes are of great value and should be listed. 

* Page 26, line 2. Are these genes particularly small causing large gaps per length of gene.

* Page 26, 8 lines from top. If genes are duplicated or have very similar sequences, then insertion reads cannot be uniquely mapped giving the appearance of being essential. 

* Page 28, Fig. 4B, purple color of the RIM pathway is hard to see.

* Page 28, 8 lines from bottom. Would be helpful here to have a comment defining the RIM pathway.

* Page 29, line 6. hypomorphs of essential genes can also be generated by c-terminal insertions in coding sequence. Do you see this? Also, this is transposon dependent. Some TEs have promoters that drive expression of adjacent genes (Michel-2016).

* Page 29, 10 lines from bottom. But it appears that insertions within the ERG11 ORF were tolerated in fluconazole. Or at least ERG11 was not placed in Fig. 4B or C. It is possible you didnt have the statistical power in the ORF. With Hermes there is much higher density of insertion in the intergenic sequences. Perhaps that is true for Ac/Ds. A metagene plot would reveal this. 

* Page 29, 7 lines from bottom. A map of the insertions upstream of ERG11 would provide visual confidence.

* Page 29, 3 lines from bottom. Is there orientation specificity to the insertions in the upstream sequence. I believe there is a cryptic promoter on one end of your Ac/Ds.

* Page 30, 8 lines from bottom. Your insertions could also be increasing expression of the mitochondrial genes. Mitchel et al found Ac/Ds has a cryptic promoter capable of expressing essential domains. To support the loss of mitochondrial function model of fluconazole resistance I would test a few deletions of mito genes?

* Page 30, 3 lines from bottom. This is an excellent use of TN-Seq maps. It would be best to illustrate this point by showing a map of the ERG11 insertions with and without drug.

* Page 31, Shiny app. This app is a great addition. I did have trouble looking up genes with three letter names. Does it only recognize systematic names? I would suggest a page of instructions. I was very interested in looking at the insertions around ERG11 but because insertions in the ORF are low its not in the app. I suggest you show the insertion patterns regardless of the statistical significance of the ORF. This will allow people to look at the promoter regions.

* Page 33, 10 lines from bottom. Typo, Here we find…

* Page 36, top line. The reader should be reminded that TN-Seq only works with haploids.

---

## [Decision Letter · Decision Letter 2]

21 Mar 2025

Dear Dr Billmyre,

Thank you for your patience while we considered your revised manuscript "Saturation transposon mutagenesis enables genome-wide identification of genes required for growth and fluconazole resistance in the human fungal pathogen Cryptococcus neoformans" for publication as a Research Article at PLOS Biology. This revised version of your manuscript has been evaluated by the PLOS Biology editors, the Academic Editor and the original reviewers.

Based on the reviews, we are likely to accept this manuscript for publication, provided you satisfactorily address the remaining editorial points. Please also make sure to address the following data and other policy-related requests.

a) We routinely suggest changes to titles to ensure maximum accessibility for a broad, non-specialist readership, and to ensure they reflect the contents of the paper. In this case, we would suggest a minor edit to the title, as follows. Please ensure you change both the manuscript file and the online submission system, as they need to match for final acceptance:

"Landscape of essential growth and fluconazole-resistance genes in the human fungal pathogen Cryptococcus neoformans"

Please supply the numerical values either in the a supplementary file or as a permanent DOI’d deposition for the following figures:

Figures 2DEF, 3A, 4BC, 5ABE, S1ABC, S3B, S4, S5

c) Please cite the location of the data clearly in all relevant main and supplementary Figure legends, e.g. “The data underlying this Figure can be found in S1 Data” or “The data underlying this Figure can be found in https://doi.org/10.5281/zenodo.XXXXX”

d) We require the original, uncropped and minimally adjusted images supporting all blot and gel results reported in the Figures S2BCD

We will require these files before a manuscript can be accepted so please prepare and upload them now. Please carefully read our guidelines for how to prepare and upload this data: https://journals.plos.org/plosbiology/s/figures#loc-blot-and-gel-reporting-requirements

e) Please provide the tree files for the phylogenetic trees in Figures 3BC

f) Many thanks for providing the underlying code in Stowers Depository. However, it is not clear if depositions here can be readily changed or deleted, so please make a permanent DOI’d copy (e.g. in Zenodo) and provide this URL in the manuscript and Data Availability Statement. I would also like to notice that the link took me to "Not found" page.

g) Please ensure that your Data Statement in the submission system accurately describes where your data can be found and is in final format, as it will be published as written there.

h) Per journal policy, if you have generated any custom code during the course of this investigation, please make it available without restrictions upon publication. Please ensure that the code is sufficiently well documented and reusable, and that your Data Statement in the Editorial Manager submission system accurately describes where your code can be found.

We expect to receive your revised manuscript within two weeks. 

*Published Peer Review History*

*Press*

Sincerely,

Melissa

Melissa Vazquez Hernandez, Ph.D.

Associate Editor

PLOS Biology

REVIEWERS' COMMENTS:

Reviewer #1: The authors have done a fine job in responding to the reviewers comments. The manuscript has been significantly improved as a consequence. I am now very happy to recommend publication.

Reviewer #2 (Henry Levin): I continue to believe this is an outstanding manuscript. I have now been convinced with the additional comments and data that it is of interest to the broad readership of PLOS Biology. I have no further concerns.

---

## [Editor Report · Decision Letter 3]

29 Apr 2025

Dear Blake,

Thank you for the submission of your revised Research Article "Landscape of essential growth and fluconazole-resistance genes in the human fungal pathogen Cryptococcus neoformans" for publication in PLOS Biology. On behalf of my colleagues and the Academic Editor, Csaba Pál, I am pleased to say that we can in principle accept your manuscript for publication, provided you address any remaining formatting and reporting issues. These will be detailed in an email you should receive within 2-3 business days from our colleagues in the journal operations team; no action is required from you until then. Please note that we will not be able to formally accept your manuscript and schedule it for publication until you have completed any requested changes.

PRESS

Sincerely, 

Melissa

Melissa Vazquez Hernandez, Ph.D., Ph.D.

Associate Editor

PLOS Biology
